# An Adaptive Entropy-Regularization Framework for Multi-Agent Reinforcement Learning

## Abstract

In this paper, we propose an adaptive entropy-regularization framework (ADER) for multi-agent reinforcement learning (RL) to learn the adequate amount of exploration for each agent based on the degree of required exploration. In order to handle instability arising from updating multiple entropy temperature parameters for multiple agents, we disentangle the soft value function into two types: one for pure reward and the other for entropy. By applying multi-agent value factorization to the disentangled value function of pure reward, we obtain a relevant metric to assess the necessary degree of exploration for each agent. Based on this metric, we propose the ADER algorithm based on maximum entropy RL, which controls the necessary level of exploration across agents over time by learning the proper target entropy for each agent. Experimental results show that the proposed scheme significantly outperforms current state-of-the-art multi-agent RL algorithms.

## 1 Introduction

RL is one of the most notable approaches to solving decision-making problems such as robot control (Hester et al., 2012; Ebert et al., 2018), traffic light control (Wei et al., 2018; Wu et al., 2020) and games (Mnih et al., 2015; Silver et al., 2017). The goal of RL is to find an optimal policy that maximizes expected return. To guarantee convergence of model-free RL, the assumption that each element in the joint state-action space should be visited infinitely often is required (Sutton & Barto, 2018), but this is impractical due to large state and/or action spaces in real-world problems. Thus, effective exploration has been a core problem in RL. In practical real-world problems, however, the given time for learning is limited and thus the learner should exploit its own policy based on its experiences so far. Hence, the learner should balance exploration and exploitation in the dimension of time and this is called *exploration-exploitation trade-off* in RL. The problem of exploration-exploitation trade-off becomes more challenging in multi-agent RL (MARL) because the state-action space grows exponentially as the number of agents increases. Furthermore, the necessity and benefit of exploration can be different across agents and even one agent's exploration can hinder other agents' exploitation. Thus, the balance of exploration and exploitation *across multiple agents* should also be considered for MARL in addition to that across the time dimension. We refer to this problem as *multi-agent exploration-exploitation trade-off*. Although there exist many algorithms for better exploration in MARL (Mahajan et al., 2019; Kim et al., 2020; Liu et al., 2021a; Zhang et al., 2021), the research on multi-agent exploration-exploitation trade-off has not been investigated much yet.

In this paper, we propose a new framework based on entropy regularization for adaptive exploration in MARL to handle the multi-agent exploration-exploitation trade-off. The proposed framework allocates different target entropy across agents and across time based on our newly-proposed metric for the benefit of further exploration for each agent. To implement the proposed framework, we adopt the method of disentanglement between exploration and exploitation (Beyer et al., 2019; Han & Sung, 2021) to decompose the joint soft value function into two types: one for the return and the other for the entropy sum. This disentanglement alleviates instability which can occur due to the updates of the temperature parameters. It also enables applying value factorization to return and entropy separately since the contribution to the reward can be different from that to the entropy from an agent's perspective. Based on this disentanglement, we propose a metric for the desired level of exploration for each agent, based on *the partial derivative of the joint value function of pure*

*return with respect to (w.r.t.) policy action entropy*. The intuition behind this choice is clear for entropy-based exploration: Agents with higher gradient of joint pure-return value w.r.t. their action entropy should increase their target action entropy resulting in higher exploration level in order to contribute more to pure return. Under the constraint of total target entropy sum across all agents, which we will impose, the target entropy of agents with lower gradient of joint pure-return value w.r.t. their action entropy will then be reduced and inclined to exploitation rather than exploration. Thus, multi-agent exploration-exploitation trade-off can be achieved. The experiments demonstrate the effectiveness of the proposed framework for multi-agent exploration-exploitation trade-off.

## 2 BACKGROUND

**Basic setup**  We consider a decentralized partially observable MDP (Dec-POMDP), which describes a fully cooperative multi-agent task (Oliehoek & Amato, 2016). Dec-POMDP is defined by a tuple $< \mathcal{N}, \mathcal{S}, \{\mathcal{A}_i\}, \mathcal{P}, \{\Omega_i\}, \mathcal{O}, r, \gamma >$, where $\mathcal{N} = \{1, 2, \cdots, N\}$ is the set of agents. At time step $t$, Agent $i \in \mathcal{N}$ makes its own observation $o_t^i \in \Omega_i$ according to the observation function $\mathcal{O}(s, i) : \mathcal{S} \times \mathcal{N} \to \Omega_i : (s_t, i) \mapsto o_t^i$, where $s_t \in \mathcal{S}$ is the global state at time step $t$. Agent $i$ selects action $a_t^i \in \mathcal{A}_i$, forming a joint action $\boldsymbol{a_t} = \{a_t^1, a_t^2, \cdots, a_t^N\}$. The joint action yields the next global state $s_{t+1}$ according to the transition probability $\mathcal{P}(\cdot|s_t, a_t)$ and a joint reward $r(s_t, a_t)$. Each agent $i$ has an observation-action history $\tau^i \in (\Omega_i \times \mathcal{A}_i)^*$ and trains its decentralized policy $\pi^i(a^i|\tau^i)$ to maximize the return $\mathbb{E}[\sum_{t=0}^{\infty} \gamma^t r_t]$. We consider the framework of centralized training with decentralized execution (CTDE), where decentralized policies are trained with additional information including the global state in a centralized way during the training phase (Oliehoek et al., 2008).

**Value Factorization**  It is difficult to learn the joint action-value function, which is defined as $Q_{JT}(s, \boldsymbol{\tau}, \boldsymbol{a}) = \mathbb{E}[\sum_{t=0}^{\infty} \gamma^t r_t | s, \boldsymbol{\tau}, \boldsymbol{a}]$ due to the problem of the curse of dimensionality as the number of agents increases. For efficient learning of the joint action-value function, *value factorization* techniques have been proposed to factorize it into individual action-value functions $Q_i(\tau^i, a^i)$, $i = 1, \cdots, N$. One representative example is QMIX, which introduces a monotonic constraint between the joint action-value function and the individual action-value function. The joint action-value function in QMIX is expressed as

$$Q_{JT}(s, \boldsymbol{\tau}, \boldsymbol{a}) = f_{mix}(s, Q_1(\tau^1, a^1), \cdots, Q_N(\tau^N, a^N)), \quad \frac{\partial Q_{JT}(s, \boldsymbol{\tau}, \boldsymbol{a})}{\partial Q_i(\tau^i, a^i)} \geq 0, \quad \forall i \in \mathcal{N}, \quad (1)$$

where $f_{mix}$ is a mixing network which combines the individual action-values into the joint action-value based on the global state. To satisfy the monotonic constraint $\partial Q_{JT}/\partial Q_i \geq 0$, the mixing network is restricted to have positive weights. There exist other value-based MARL algorithms with value factorization (Son et al., 2019; Wang et al., 2020a). Actor-critic based MARL algorithms also considered value factorization to learn the centralized critic (Peng et al., 2021; Su et al., 2021).

**Maximum Entropy RL and Entropy Regularization**  Maximum entropy RL aims to promote exploration by finding an optimal policy that maximizes the sum of cumulative reward and entropy (Haarnoja et al., 2017; 2018a). The objective function of maximum entropy RL is given by

$$J_{MaxEnt}(\pi) = \mathbb{E}_\pi \left[ \sum_{t=0}^{\infty} \gamma^t (r_t + \alpha \mathcal{H}(\pi(\cdot|s_t))) \right], \quad (2)$$

where $\mathcal{H}(\cdot)$ is the entropy function and $\alpha$ is the temperature parameter which determines the importance of the entropy compared to the reward. Soft actor-critic (SAC) is an off-policy actor-critic algorithm which efficiently solves the maximum entropy RL problem (2) based on soft policy iteration, which consists of soft policy evaluation and soft policy improvement. For this, the soft Q function is defined as the sum of the total reward and the future entropy, i.e., $Q^\pi(s_t, a_t) := r_t + \mathbb{E}_{\tau_{t+1} \sim \pi} \left[ \sum_{l=t+1}^{\infty} \gamma^{l-t}(r_l + \sum_{i=1}^{N} \alpha \mathcal{H}(\pi(\cdot|s_l))) \right]$. In the soft policy evaluation step, for a fixed policy $\pi$, the soft Q function is estimated with convergence guarantee by repeatedly applying the soft Bellman backup operator $\mathcal{T}_{sac}^\pi$ to an estimate function $Q : \mathcal{S} \times \mathcal{A} \to \mathbb{R}$, and the soft Bellman backup operator is given by $\mathcal{T}_{sac}^\pi Q(s_t, a_t) = r_t + \gamma \mathbb{E}_{s_{t+1}}[V(s_{t+1})]$, where $V(s_t) = \mathbb{E}_{a_t \sim \pi}[Q(s_t, a_t) - \alpha \log \pi(a_t|s_t)]$. In the soft policy improvement step, the policy is updated using the evaluated soft Q function as follows: $\pi_{new} = \arg \max_\pi \mathbb{E}_{a_t \sim \pi} [Q^{\pi_{old}}(s_t, a_t) - \alpha \log \pi(a_t|s_t)]$. By iterating the soft policy evaluation and soft policy improvement, called the soft policy iteration,

SAC converges to an optimal policy that maximizes (2) within the considered policy class in the case of finite MDPs. SAC also works effectively for large MDPs with function approximation.

One issue with SAC is the adjustment of the hyperparameter $\alpha$ in (2), which control the relative importance of the entropy with respect to the reward. The magnitude of the reward depends not only on tasks but also on the policy which improves over time during the training phase. Thus, Haarnoja et al. (2018b) proposed a method to adjust the temperature parameter $\alpha$ over time to guarantee the minimum average entropy at each time step. For this, they reformulated the maximum entropy RL as the following entropy-regularized optimization:

$$J_{ER}(\pi_{0:T}) = \mathbb{E}_{\pi_{0:T}} \left[ \sum_{t=0}^{T} r_t \right] \quad \text{s.t. } \mathbb{E}_{(s_t,a_t) \sim \pi_t} \left[ -\log(\pi_t(a_t|s_t)) \right] \geq \mathcal{H}_0 \tag{3}$$

where $\mathcal{H}_0$ is the target entropy. Here, to optimize the objective (3), the technique of dynamic programming is used, i.e., $\max_{\pi_{t:T}} \mathbb{E}[\sum_{i=t}^{T} r_i] = \max_{\pi_t} \left\{ \mathbb{E}[r_t] + \max_{\pi_{t+1:T}} \mathbb{E}[\sum_{i=t+1}^{T} r_i] \right\}$. Starting from time step $T$, we obtain the optimal policy $\pi_{0:T}^*$ and $\alpha_{0:T}^*$ by applying the backward recursion. That is, we begin with the constrained optimization at time step $T$, given by

$$\max_{\pi_T} \mathbb{E}[r_T] \quad \text{s.t. } \mathbb{E}_{(s_T,a_T) \sim \pi_T} \left[ -\log(\pi_T(a_T|s_T)) \right] \geq \mathcal{H}_0 \tag{4}$$

and convert the problem into the Lagrangian dual problem as follows:

$$\min_{\alpha_T} \max_{\pi_T} \mathbb{E}[r_T - \alpha_T \log \pi_T(a_T|s_T)] - \alpha_T \mathcal{H}_0 = \min_{\alpha_T} \mathbb{E}[-\alpha_T \log \pi_T^*(a_T|s_T) - \alpha_T \mathcal{H}_0]. \tag{5}$$

Here, the optimal temperature parameter $\alpha_T^*$ at time step $T$, which corresponds to the Lagrangian multiplier, is obtained by solving the problem (5). Then, the backward recursion can be applied to obtain optimal $\alpha$ at time step $t$ based on the Lagrange dual problem:

$$\alpha_t^* = \arg\min_{\alpha_t} \underbrace{\mathbb{E}_{a_t \sim \pi_t^*}[-\alpha_t \log \pi_t^*(a_t|s_t) - \alpha_t \mathcal{H}_0]}_{:=J(\alpha_t)}, \tag{6}$$

where $\pi_t^*$ is the maximum entropy policy at time step $t$. Here, by minimizing the loss function $J(\alpha)$, $\alpha$ is updated to increase (or decrease) if the entropy of policy is lower (or higher) than the target entropy. In the infinite-horizon case, the discount factor $\gamma$ is included and $\pi_t^*$ is replaced with the current approximate maximum entropy solution by SAC. In this way, the soft policy iteration of SAC is combined with the $\alpha$ adjustment based on the loss function $J(\alpha)$ defined in (6). This algorithm effectively handles the reward magnitude change over time during training (Haarnoja et al., 2018b). Hence, one needs to set only the target entropy $\mathcal{H}_0$ for each task and then $\alpha$ is automatically adjusted over time for the target entropy.

**Related Works** Here, we mainly focus on the entropy-based MARL. Other related works regarding multi-agent exploration are provided in Appendix E. There exist previous works on entropy-based MARL. Zhou et al. (2020) proposed an actor-critic algorithm, named LICA, which learns implicit credit assignment and regularizes the action entropy by dynamically controlling the magnitude of the gradient regarding entropy to address the high sensitivity of the temperature parameter caused by the curvature of derivative of entropy. LICA allows multiple agents to perform consistent level of exploration. However, LICA does not maximize the cumulative sum of entropy but regularize the action entropy. Zhang et al. (2021) proposed an entropy-regularized MARL algorithm, named FOP, which introduces a constraint that the entropy-regularized optimal joint policy is decomposed into the product of the optimal individual policies. FOP introduced a weight network to determine individual temperature parameters. Zhang et al. (2021) considered individual temperature parameters for updating policy, but in practice, they used the same value (for all agents) which is annealed during training for the temperature parameters. This encourages multiple agents to focus on exploration at the beginning of training, which considers exploration-exploitation only in time dimension in a heuristic way.

A key point is that the aforementioned algorithms maximize or regularize the entropy of the policies to encourage *the same level of exploration across the agents*. Such exploration is still useful for several benchmark tasks but cannot handle the multi-agent exploration-exploitation trade-off. Furthermore, in the previous methods, the joint soft Q-function defined as the total sum of return and entropy is directly factorized by value decomposition, and hence the return is not separated from the entropy in the Q-value. From the perspective of one agent, however, the contribution to the reward and that to the entropy can be different. What we actually need to assess the goodness of a policy is the return estimate, which is difficult to obtain by such unseparated factorization.

# 3 METHODOLOGY

In order to address the aforementioned problems, we propose an **AD**aptive **E**ntropy-**R**egularization framework (ADER), which can balance *exploration and exploitation across multiple agents* by learning the target entropy for each agent.

## 3.1 MOTIVATION

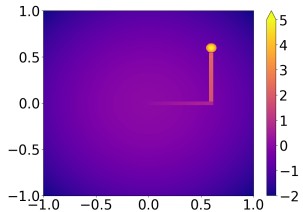

Figure 1: Reward surface in the considered matrix game. $a_1$ and $a_2$ correspond to x-axis and y-axis, respectively.

The convergence of model-free RL requires the assumption that all state-action pairs should be visited infinitely often, and this necessitates exploration (Sutton & Barto, 2018). In practice, however, the number of time steps during which an agent can interact with the environment is limited. Thus, a balance between exploration and exploitation in the dimension of time is crucial for high performance in RL. Furthermore, in the case of MARL, a balance between exploration and exploitation in the dimension of agents should be considered. This is because 1) the degree of necessity and benefit of exploration can be different across multiple agents and 2) one agent's exploration can hinder other agents' exploitation, resulting in the situation that simultaneous exploration of multiple agents can make learning unstable. We refer to this problem as *multi-agent exploration-exploitation trade-off*. To handle the problem of multi-agent exploration-exploitation trade-off, we need to control the amount of exploration of each agent adaptively and learn this amount across agents (i.e., agent dimension) and over time (i.e., time dimension). In the case of entropy-based exploration, we should allocate higher target entropy values to the agents who need more exploration or have larger benefit from exploration and allocate lower target entropy values to the agents who need more exploitation or have less benefit from exploration. In order to see the necessity of such adaptive exploration-exploitation trade-off control in MARL, let us consider a modified continuous cooperative matrix game (Peng et al., 2021). The considered game consists of two agents: each agent has an one-dimensional continuous action $a^i$ which is bounded in $[-1, 1]$. The shared reward is determined by the joint action, and the reward surface is given in Fig. 1. As seen in Fig. 1, there is a connected narrow path from the origin $(0, 0)$ to $(0.6, 0.55)$, consisting of two subpaths: one from $(0, 0)$ to $(0.6, 0)$ and the other from $(0.6, 0)$ to $(0.6, 0.55)$. There is a circle with center at $(0.6, 0.6)$ and radius $0.05$. The reward gradually increases only along the path as the position approaches the center of the circle and the maximum reward is $5$. There is a penalty if the joint action yields the position outside the path or the circle, and the penalty value increases as the outside position is farther from the origin $(0, 0)$. The agents start from the origin with initial action pair $\boldsymbol{a} = (0, 0)$ and want to learn to reach the circle along the path. Even if this game is stateless, exploration for action space is required to find the action $(0.6, 0.6)$. One can think that one can find the optimal joint action once the action near the circle is selected. However, the action starting with $(0,0)$ cannot jump to $(0.6, 0.6)$ since we use function approximators for the policies and train them based on stochastic gradient descent. The action should be trained to reach the circle along the two subpaths. In the beginning, to go through the first subpath, $a_2$ (i.e., $y$-axis movement) should not fluctuate from $0$ and $a_1$ should be trained to increase upto $0.6$. In this phase, if $a_2$ explores too much, the positive reward is rarely obtained. Then, $a_1$ is not trained to increase upto $0.6$ because of the penalty. Once the joint action is trained to $(0.6, 0)$, on the other hand, the necessity of exploration is changed. In this phase, $a_1$ should keep its action at $0.6$, whereas $a_2$ should be trained to increase upto $0.55$. As seen in this example, it is important to control the trade-off between exploitation and exploration across multiple agents. In addition, we should update the trade-off over time because the required trade-off can change during the learning process. As we will see in Section 4, a method that retains the same or different-but-constant level of exploration across all agents fails to learn in this continuous cooperative matrix game. Thus, we need a framework that can adaptively learn appropriate levels of exploration for all agents over time, considering the time-varying multi-agent exploration-exploitation trade-off.

## 3.2 ADAPTIVE ENTROPY-REGULARIZED MARL

We now propose our ADER framework enabling adaptive exploration capturing the multi-agent exploration-exploitation trade-off. One can adopt the entropy constrained objective defined in (3) and extend it to multi-agent systems. A simple extension is to maximize the team reward while keeping

the average entropy of each agent above the same target entropy. For the sake of convenience, we call this scheme simple entropy-regularization for MARL (SER-MARL). However, SER-MARL cannot handle the multi-agent exploration-exploitation trade-off because the amounts of exploration for all agents are the same. One can also consider different but fixed target entropies for multiple agents. However, this case cannot handle the time-varying behavior of multi-agent exploitation-exploration trade-off, discussed in the previous subsection with Fig. 1. Thus, to incorporate the multi-agent exploration-exploitation trade-off, we consider the following optimization problem:

$$\max_{\boldsymbol{\pi}} \mathbb{E}_{\boldsymbol{\pi}} \left[ \sum_{t=0}^{\infty} \gamma^t r_t \right] \quad \text{s.t. } \mathbb{E}_{(s_t, \boldsymbol{a_t}) \sim \boldsymbol{\pi}} \left[ -\log(\pi_t^i(a_t^i | \tau_t^i)) \right] \geq \mathcal{H}_i, \ \forall i \in \mathcal{N} \text{ and } \sum_{j=1}^{N} \mathcal{H}_j = \mathcal{H}_0, \quad (7)$$

where $\boldsymbol{\pi} = (\pi^1, \cdots, \pi^N)$, $\mathcal{H}_i$ is the target entropy of Agent $i$, and $\mathcal{H}_0$ is the total sum of all target entropies. The key point here is that we fix the target entropy sum as $\mathcal{H}_0$ but each $\mathcal{H}_i$ is adaptive and learned. The total entropy budget $\mathcal{H}_0$ is shared by all agents. When some agents' target entropy values are high for more exploration, the target entropy values of other agents should be low, leading to more exploitation, due to the fixed total entropy budget. Thus, the exploitation-exploration trade-off across agents (i.e., agent dimension) can be captured. The main challenge is how to learn individual target entropy values $\mathcal{H}_1, \cdots, \mathcal{H}_N$ over time (i.e., time dimension) as the learning progresses.

We postpone the presentation of our method of learning the individual target entropy values to Section 3.4. Here, we consider how to solve the problem (7) when $\mathcal{H}_1, \cdots, \mathcal{H}_N$ are determined. In order to solve the problem (7), one can simply extend the method in (Haarnoja et al., 2018b) to the MARL case. That is, one can first consider a finite-horizon case with terminal time step $T$, apply approximate dynamic programming and the Lagrange multiplier method, obtain the update formula at time step $t$, and then relax to the infinite-horizon case by introducing the discount factor, as in (Haarnoja et al., 2018b). For this, the joint soft Q-function $Q_{JT}(s_t, \boldsymbol{\tau_t}, \boldsymbol{a_t})$ can be defined as

$$Q_{JT}(s_t, \boldsymbol{\tau_t}, \boldsymbol{a_t}) := r_t + \mathbb{E}_{\tau_{t+1} \sim \pi} \left[ \sum_{l=t+1}^{\infty} \gamma^{l-t} (r_l + \sum_{i=1}^{N} \alpha^i \mathcal{H}(\pi^i(\cdot | \tau_l^i))) \right], \quad (8)$$

and then this joint soft Q-function is estimated based on the following Bellman backup operator: $\mathcal{T}^{\pi} Q_{JT}(\boldsymbol{\tau_t}, \boldsymbol{a_t}) := r_t + \gamma \mathbb{E}_{\tau_{t+1}} [V(s_{t+1}, \boldsymbol{\tau_{t+1}})]$, where $V_{JT}(s_t, \boldsymbol{\tau_t}) = \mathbb{E}_{a_t \sim \boldsymbol{\pi}} \left[ Q_{JT}(s_t, \boldsymbol{\tau_t}, \boldsymbol{a_t}) - \sum_{i=1}^{N} \alpha^i \log \pi(a_t^i | \tau_t^i) \right]$. However, optimizing the objective (7) based on the joint soft Q-function in (8) and the corresponding Bellman operator $\mathcal{T}^{\pi}$ has several limitations. First, the estimation of the joint soft Q-function can be unstable due to the changing $\{\alpha^i\}_{i=1}^{N}$ in (8) as the determined target entropy values are updated over time. Second, we cannot apply value factorization to return and entropy separately because the joint soft Q-function defined in (8) estimates only the sum of return and entropy. For a single agent, the contribution to the global reward may be different from that to the total entropy. Thus, learning to decompose the entropy can prevent the mixing network from learning to decompose the global reward. Furthermore, due to the inseparability of reward and entropy, it is difficult to pinpoint each agent's contribution sensitivity to the global reward itself, which is used for assessing the necessity and benefit of more exploration.

## 3.3 Disentangled Exploration and Exploitation

To address the aforementioned problems and facilitate the acquisition of a metric for the degree of required exploration for each agent in MARL, we disentangle the return from the entropy by decomposing the joint soft Q-function into two types of Q-functions: One for reward and the other for entropy. That is, the joint soft Q-function is decomposed as $Q_{JT}(\boldsymbol{\tau_t}, \boldsymbol{a_t}) = Q_{JT}^R(\boldsymbol{\tau_t}, \boldsymbol{a_t}) + \sum_{i=1}^{N} \alpha^i Q_{JT}^{H,i}(\boldsymbol{\tau_t}, \boldsymbol{a_t})$, where $Q_{JT}^R(\boldsymbol{\tau_t}, \boldsymbol{a_t})$ and $Q_{JT}^{H,i}(\boldsymbol{\tau_t}, \boldsymbol{a_t})$ are the joint action value function for reward and the joint action value function for the entropy of Agent $i$'s policy, respectively, given by

$$Q_{JT}^R(s_t, \boldsymbol{\tau_t}, \boldsymbol{a_t}) = r_t + \mathbb{E}_{\tau_{t+1} \sim \boldsymbol{\pi}} \left[ \sum_{l=t+1}^{\infty} \gamma^{l-t} r_l \right] \quad \text{and} \quad (9)$$

$$Q_{JT}^{H,i}(s_t, \boldsymbol{\tau_t}, \boldsymbol{a_t}) = \mathbb{E}_{\boldsymbol{\tau_{t+1}} \sim \boldsymbol{\pi}} \left[ \sum_{l=t+1}^{\infty} \gamma^{l-t} \mathcal{H}(\pi^i(\cdot | \tau_t^i)) \right], \quad i \in \mathcal{N}. \quad (10)$$

The action value functions $Q_{JT}^R(s_t, \boldsymbol{\tau_t}, \boldsymbol{a_t})$ and $Q_{JT}^{H,i}(s_t, \boldsymbol{\tau_t}, \boldsymbol{a_t})$ can be estimated based on their corresponding Bellman backup operators, defined by

$$\mathcal{T}_R^\pi Q_{JT}^R(s_t, \boldsymbol{\tau_t}, \boldsymbol{a_t}) := r_t + \gamma \mathbb{E}\left[V_{JT}^R(s_t, \boldsymbol{\tau_{t+1}})\right], \quad \mathcal{T}_{H,i}^\pi Q_{JT}^{H,i}(s_t, \boldsymbol{\tau_t}, \boldsymbol{a_t}) := \gamma \mathbb{E}\left[V_{JT}^{H,i}(s_t, \boldsymbol{\tau_{t+1}})\right] \tag{11}$$

where $V_{JT}^R(s_t, \boldsymbol{\tau_t}) = \mathbb{E}_{\boldsymbol{a_t}}\left[Q_{JT}^R(s_t, \boldsymbol{\tau_t}, \boldsymbol{a_t})\right]$ and $V_{JT}^{H,i}(s_t, \boldsymbol{\tau_t}) = \mathbb{E}_{\boldsymbol{a_t}}\left[Q_{JT}^{H,i}(s_t, \boldsymbol{\tau_t}, \boldsymbol{a_t})\right.$ $\left. - \alpha^i \log \pi(a_t^i | \tau_t^i)\right]$ are the joint value functions regarding reward and entropy, respectively.

**Proposition 1** *The disentangled Bellman operators $\mathcal{T}_R^\pi$ and $\mathcal{T}_{H,i}^\pi$ are contractions.*

*Proof:* See Appendix A.

Now we apply value decomposition using a mixing network (Rashid et al., 2018) to represent each of the disentangled joint action-value and value functions as a mixture of individual value functions. For instance, the joint value function for reward $V_{JT}^R(s, \boldsymbol{\tau})$ is decomposed as $V_{JT}^R(s, \boldsymbol{\tau}) = f_{mix}^{V,R}(s, V_1^R(\tau^1), \cdots, V_N^R(\tau^N))$, where $V_i^R(\tau^i)$ is the individual value function of Agent $i$ and $f_{mix}^{V,R}$ is the mixing network for the joint value function for reward. Similarly, we apply value decomposition and mixing networks to $Q_{JT}^R(\boldsymbol{\tau_t}, \boldsymbol{a_t})$ and $Q_{JT}^{H,i}(\boldsymbol{\tau_t}, \boldsymbol{a_t})$, $i \in \mathcal{N}$.

Based on the disentangled joint soft Q-functions, the optimal policy and the temperature parameters can be obtained as functions of $\mathcal{H}_1, \cdots, \mathcal{H}_N$ by using a similar technique to that in (Haarnoja et al., 2018b) based on dynamic programming and Lagrange multiplier. That is, we first consider the finite-horizon case and apply dynamic programming with backward recursion: $\max_{\boldsymbol{\pi_{t:T}}} \mathbb{E}\left[\sum_{i=t}^T r_i\right] =$

$$\max_{\pi_t}\left(\mathbb{E}[r_t] + \max_{\pi_{t+1:T}}\left(\mathbb{E}[\sum_{i=t+1}^T r_i],\right)\right) \text{ s.t. } \mathbb{E}_{(s_t, \boldsymbol{a_t}) \sim \boldsymbol{\pi_t}}\left[-\log(\pi_t^i(a_t^i | \tau_t^i))\right] \geq \mathcal{H}_i, \ \forall t, i. \tag{12}$$

We can obtain the optimal policy and the temperature parameters by recursively solving the dual problem from the last time step $T$ by using the technique of Lagrange multiplier. At time step $t$, the optimal policy is obtained for given temperature parameters, and the optimal temperature parameters are computed based on the obtained optimal policy as follows:

$$\boldsymbol{\pi_t^*} = \arg\max_{\boldsymbol{\pi_t}} \mathbb{E}_{\boldsymbol{a_t} \sim \boldsymbol{\pi_t}}\left[\underbrace{Q_{JT}^{R*}(s_t, \boldsymbol{\tau_t}, \boldsymbol{a_t})}_{(a)} + \sum_{i=1}^N \alpha_t^i \underbrace{(Q_{JT}^{H*,i}(s_t, \boldsymbol{\tau_t}, \boldsymbol{a_t}) - \log\boldsymbol{\pi_t}(a_t^i | \tau_t^i))}_{(b)}\right] \tag{13}$$

$$\alpha_t^{i*} = \arg\min_{\alpha_t^i} \mathbb{E}_{\boldsymbol{a_t} \sim \boldsymbol{\pi_t^*}}\left[-\alpha_t^i \log\boldsymbol{\pi_t^*}(a_t^i | \tau_t^i) - \alpha_t^i \mathcal{H}_i\right], \quad \forall i \in \mathcal{N}. \tag{14}$$

In the infinite-horizon case, (13) and (14) provide the update formulae at time step $t$, and the optimal policy is replaced with the current approximate multi-agent maximum-entropy solution, which can be obtained by extending SAC to MARL. Note that maximizing the term (a) in (13) corresponds to the ultimate goal of MARL, i.e., the expected return. On the other hand, maximizing the term (b) in (13) corresponds to enhancing exploration of Agent $i$.

### 3.4 Learning Individual Target Entropy Values

In the formulation (7), the amount of exploration for Agent $i$ is controlled by the target entropy $\mathcal{H}_i$ under the sum constraint $\sum_{j=1}^N \mathcal{H}_j = \mathcal{H}_0$. In this subsection, we describe how to determine the target entropy for each agent over time. First, we represent the target entropy of Agent $i$ as $\mathcal{H}_i = \beta_i \times \mathcal{H}_0$ with $\sum_{i=1}^N \beta_i = 1$ to satisfy the entropy sum constraint. Then, we need to learn $\beta_i$ over time $t$. Considering the fact that the ultimate goal is to maximize the return and this is captured by the value function of return $V_{JT}^R$ by disentanglement, we adopt the *partial derivative $\partial V_{JT}^R / \partial \mathcal{H}(\pi_t^i)$ at time $t$* to assess the benefit of increasing the target entropy $\mathcal{H}_i$ of Agent $i$ for more exploration at time $t$. Note that $\partial V_{JT}^R / \partial \mathcal{H}(\pi_t^i)$ denotes the change in the joint pure-return value w.r.t. the differential increase in Agent $i$'s policy action entropy. Suppose that $\partial V_{JT}^R / \partial \mathcal{H}(\pi_t^i) > \partial V_{JT}^R / \partial \mathcal{H}(\pi_t^j)$ for two agents $i$ and $j$. Then, if we update two policies $\pi_t^i$ and $\pi_t^j$ to two new policies so that the entropy of each of the two policies is increased by the same amount $\Delta\mathcal{H}$, then Agent $i$ contributes more to the

pure return than Agent $j$. Then, under the total entropy sum constraint, the targe entropy of Agent $i$ should be assigned higher than that of Agent $j$ for higher return. Furthermore, when this quantity for a certain agent is largely negative, increasing the target entropy for this agent can decrease the joint (return) value significantly, which implies that exploration of this agent can hinder other agents' exploitation. Therefore, we allocate higher (or lower) target entropy to agents whose $\partial V_{JT}^R / \partial \mathcal{H}(\pi_t^i)$ is larger (or smaller) than that of other agents. With the proposed metric, in the case of $\mathcal{H}_0 \geq 0$, we set the coefficients $\beta_i$, $i = 1, \cdots, N$ for determining the individual target entropy values $\mathcal{H}_i$, $i = 1, \cdots, N$ as follows: $\boldsymbol{\beta} = \begin{bmatrix} \beta_1, \cdots, \beta_i, \cdots, \beta_N \end{bmatrix} =$

$$\text{Softmax}\left[ \mathbb{E}\left[ \frac{\partial V_{JT}^R(s, \boldsymbol{\tau})}{\partial \mathcal{H}(\pi_t^1(\cdot|\tau^1))} \right], \cdots, \mathbb{E}\left[ \frac{\partial V_{JT}^R(s, \boldsymbol{\tau})}{\partial \mathcal{H}(\pi_t^i(\cdot|\tau^i))} \right], \cdots, \mathbb{E}\left[ \frac{\partial V_{JT}^R(s, \boldsymbol{\tau})}{\partial \mathcal{H}(\pi_t^N(\cdot|\tau^N))} \right] \right]. \tag{15}$$

The relative required level of exploration across agents can change as the learning process and this is captured in these partial derivatives. We compute the partial derivative for (15) in continuous and discrete action cases as follows:

$$\frac{\partial V_{JT}^R(s, \boldsymbol{\tau})}{\partial \mathcal{H}(\pi_t^i(\cdot|\tau^i))} = \begin{cases} \frac{\partial V_{JT}^R(s, \boldsymbol{\tau})}{\partial \log \sigma^i(\tau^i)}, & \text{Gaussian policy for continuous action} \\ \frac{\partial V_{JT}^R(s, \boldsymbol{\tau})}{\partial V_i^R(\tau^i)} \times \frac{\partial V_i^R(\tau^i)}{\partial \mathcal{H}(\pi_t^i)}, & \text{Categorical policy for discrete action} \end{cases}, \tag{16}$$

where $\sigma^i$ is the standard deviation of Agent $i$'s Gaussian policy. In the case of Gaussian policy for continuous action, the partial derivative $\partial V_{JT}^R / \partial \mathcal{H}(\pi^i)$ is obtained as the partial derivative w.r.t. the log of the standard deviation of Gaussian policy based on the fact that the entropy of Gaussian random variable with variance $\sigma_i^2$ is $\log(\sqrt{2\pi e}\sigma_i)$. This can be done by adopting the reparameterization trick. On the other hand, it is difficult to directly obtain the partial derivative in the discrete-action case based on the categorical policy. For this, we use the chain rule to compute the partial derivative $\partial V_{JT}^R / \partial \mathcal{H}(\pi_t^i)$ as shown in (16), where we numerically compute $\frac{\partial V_i^R(\tau^i)}{\partial \mathcal{H}(\pi_t^i)} \approx \frac{\Delta V_i^R(\tau^i)}{\Delta \mathcal{H}(\pi_t^i)}$. For numerical computation, we first update the policy in the direction of maximizing entropy and then compute the changes of $V_i^R(\tau^i)$ and $\mathcal{H}(\pi_t^i)$ to obtain the approximation. That is, the approximation is given by $\frac{\Delta V_i^R(\tau^i)}{\Delta \mathcal{H}(\pi_t^i)} = \frac{V_i^R(\tau^i; \pi'^i_t) - V_i^R(\tau^i; \pi_t^i)}{\mathcal{H}(\pi'^i_t) - \mathcal{H}(\pi_t^i)}$ by updating $\pi_t^i$ to $\pi'^i_t$ in direction of maximizing $\mathcal{H}(\pi_t^i)$. A detailed explanation of computation of the metric is provided in Appendix B.1.

During the training phase, we continuously compute (15) from the samples in the replay buffer and set the target entropy values. Instead of using the computed values directly, we apply exponential moving average (EMA) filtering for smoothing. The exponential moving average filter prevents the target entropy from changing abruptly. More concretely, if the partial derivative $\partial V_{JT}^R / \partial \mathcal{H}(\pi_t^i)$ has a large variance over the samples in the replay buffer, the computed metric can fluctuate whenever the transitions are sampled. This causes instability in learning, and thus the EMA filter can prevent the instability by smoothing the value. The output of EMA filter $\boldsymbol{\beta}^{EMA} = \begin{bmatrix} \beta_1^{EMA}, \cdots, \beta_N^{EMA} \end{bmatrix}$ is computed recursively as

$$\boldsymbol{\beta}^{EMA} \leftarrow (1 - \xi)\boldsymbol{\beta}^{EMA} + \xi\boldsymbol{\beta} \tag{17}$$

where $\boldsymbol{\beta}$ is given in (15) and $\xi \in [0, 1]$. Thus, the target entropy is given by $\mathcal{H}_i = \beta_i^{EMA} \times \mathcal{H}_0$.

Finally, the procedure of ADER is composed of the policy evaluation based on the Bellman operators and Proposition 1, the policy update for policy and temperature parameters in (13) and (14), and the target entropy update in (15) and (17). The detailed implementation is provided in Appendix B.

## 4 EXPERIMENTS

In this section, we provide numerical results and ablation studies. We first present the result on the continuous matrix game described in Sec. 3.1 and then results including sparse StarCraft II micromanagement (SMAC) tasks (Samvelyan et al., 2019).

**Continuous Cooperative Matrix Game**  As mentioned in Sec.3.1, the goal of this environment is to learn two actions $a_1$ and $a_2$ so that the position $(a_1, a_2)$ starting from $(0, 0)$ to reach the target circle along a narrow path, as shown in Fig. 1. The maximum reward 5 is obtained if the position reaches the center of the circle. We compare ADER with four baselines. One is SER-MARL with the same target entropy for all agents. The second is SER-MARL with different-but-constant target entropy values for two agents (SER-DCE). Here, we set a higher target entropy for $a_1$ than $a_2$. The

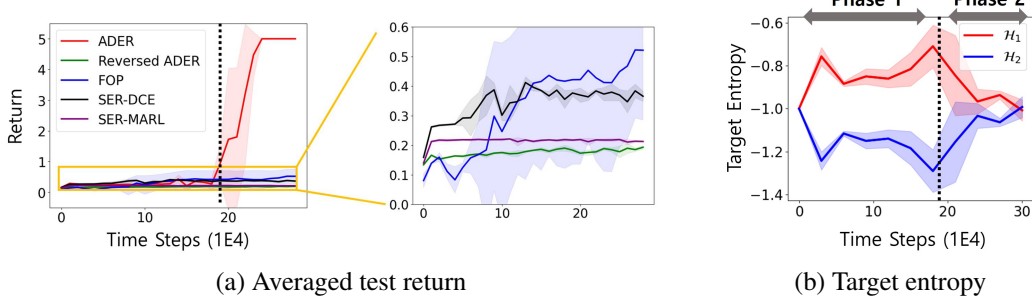

(a) Averaged test return         (b) Target entropy

Figure 2: (a) The performance of ADER and the baselines on the considered matrix game (The performance of the baseline marked with a yellow box is enlarged and displayed, and the black dotted line denotes the time when the position reaches the junction of the two subpaths) and (b) The learned target entropy values during the training.

third is Reversed ADER, which reversely uses the proposed metric $-\partial V_{jT}^{R}/\partial \mathcal{H}(\pi_t^i)$ for the level of required exploration. The fourth is FOP, which is an entropy-regularized MARL algorithm.

Fig. 2(a) shows the performance of ADER and the baselines averaged over 5 random seeds. It is seen that the considered baselines fail to learn to reach the target circle, whereas ADER successfully learns to reach the circle. Here, the different-but-constant target entropy values of SER-DCE are fixed as $(\mathcal{H}_1, \mathcal{H}_2) = (-0.7, -1.3)$, which are the maximum entropy values in ADER. It is observed that SER-DCE performs slightly better than SER-MARL but cannot learn the task with time-varying multi-agent exploration-exploitation trade-off. Fig. 2(b) shows the target entropy values $\mathcal{H}_1$ and $\mathcal{H}_2$ for $a_1$ and $a_2$, respectively, which are learned with the proposed metric during training, and shows how ADER learns to reach the target circle based on adaptive exploration. The black dotted lines in Figs. 2(a) and (b) denote the time when the position reaches the junction of the two subpaths. Before the dotted line (phase 1), ADER learns so that the target entropy of $a_1$ increases whereas the target entropy of $a_2$ decreases. So, Agent 1 and Agent 2 are trained so as to focus on exploration and exploitation, respectively. After the black dotted line (phase 2), the learning behaviors of target entropy values of $a_1$ and $a_2$ are reversed so that Agent 1 now does exploitation and Agent 2 does exploration. That is, the trade-off of exploitation and exploration is changed across the two agents. In the considered game, ADER successfully learns the time-varying trade-off of multi-agent exploration-exploitation by learning appropriate target entropies for all agents.

**Continuous Action Tasks** We evaluated ADER on two complex continuous action tasks: multi-agent HalfCheetah (Peng et al., 2021) and heterogeneous predator-prey (H-PP). The multi-agent HalfCheetah divides the body into disjoint sub-graphs and each sub-graph corresponds to an agent. We used $6 \times$ 1-HalfCheetah, which consists of six agents with one action dimension. Next, the H-PP consists of three agents, where the maximum speeds of an agent and other agents are different. In both environments, each agent has a different role

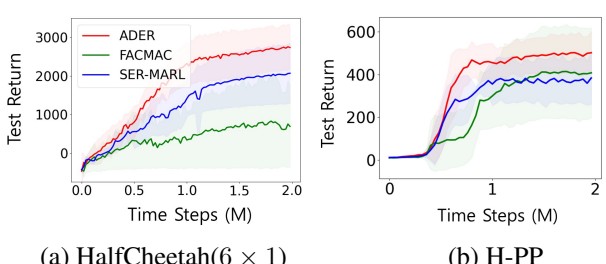

(a) HalfCheetah($6 \times 1$)    (b) H-PP

Figure 3: Comparison of ADER with SER-MARL and FACMAC on multi-agent HalfCheetah and H-PP

to achieve the common goal and thus the multi-agent exploration-exploitation tradeoff should be considered. Here, we used two baselines: SER-MARL and FACMAC Peng et al. (2021). In Fig. 3 showing the performances of ADER and the baselines averaged over 9 random seeds, ADER outperforms the considered baselines.

**Starcraft II** We also evaluated ADER on the StarcraftII micromanagement benchmark (SMAC) environment (Samvelyan et al., 2019). To make the problem more difficult, we modified the SMAC environment to be sparse. The considered sparse reward setting consisted of a dead reward and time-penalty reward. The dead reward was given only when an ally or an enemy died. Unlike the original reward in SMAC which gives the hit-point damage dealt as a reward, multiple agents did not

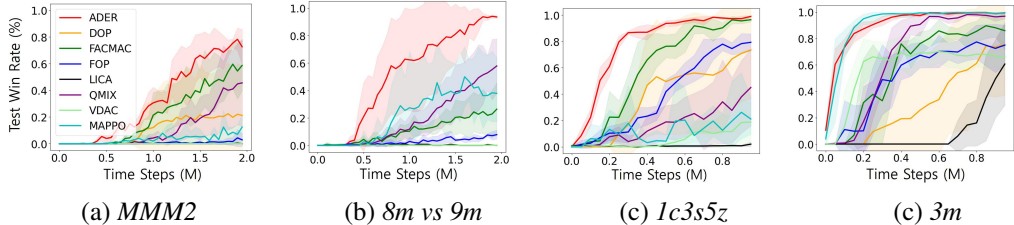

(a) *MMM2*    (b) *8m vs 9m*    (c) *1c3s5z*    (c) *3m*

Figure 4: Average test win rate on the SMAC maps. More results are provided in Appendix D. receive a reward for damaging the enemy immediately in our sparse reward setting. We compared ADER with six state-of-the-art baselines: DOP (Wang et al., 2020b), FACMAC (Peng et al., 2021), FOP (Zhang et al., 2021), LICA (Zhou et al., 2020), QMIX (Rashid et al., 2018), VDAC(Su et al., 2021) and MAPPO (Yu et al., 2021). For evaluation, we conducted experiments on the different SMAC maps with 5 different random seeds. Fig. 4 shows the performance of ADER and the considered seven baselines on the modified SMAC environment. It is seen that ADER significantly outperforms other baselines in terms of training speed and final performance. Especially in the hard tasks with imbalance between allies and enemies such as *MMM2*, and *8m vs 9m*, it is difficult to obtain a reward due to the simultaneous exploration of multiple agents. Thus, consideration of multi-agent exploration-exploitation trade-off is required to solve the task, and it seems that ADER effectively achieves this goal.

We additionally provide several experiments on the original SMAC tasks and Google Research Football (GRF) task in Appendix D.

**Ablation Study** We provide an analysis of learning target entropy in the continuous cooperative matrix game. Through the analysis, we can see how the changing target entropy affects the learning as seen in Fig. 2. In addition, we conducted an ablation study on the key factors of ADER in the SMAC environment. First, we compared ADER with SER-MARL. As in the continuous action tasks, Fig. 5 shows that ADER outperforms SER-MARL. From the result, it is seen that consideration of the multi-agent exploration-exploitation trade-off yields better performance. Sec-

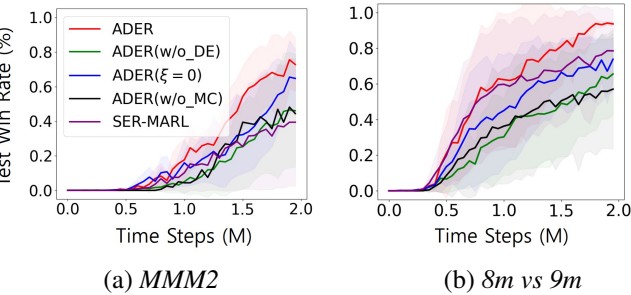

(a) *MMM2*    (b) *8m vs 9m*

Figure 5: Ablation study: Disentangled exploration (DE), EMA filter ($\xi = 0$), SER-MARL (fixed target entropy) and the monotonic constraint (MC)

ond, we compared ADER with and without the EMA filter. As seen in Fig. 5, it seems that the EMA filter enhances the stability of ADER. Third, we conducted an experiment to access the effectiveness of disentangling exploration and exploitation. We implemented ADER based on one critic which estimates the sum of return and entropy. As seen in Fig. 5, using two types of value functions yields better performance. Lastly, we compare ADER with and without the monotonic constraint to show the necessity of the monotonic constraint. It is seen that enforcing the constraint improves performance. We provided the training details for all considered environments and further ablation studies in Appendix C and D, respectively.

## 5    CONCLUSION

We have proposed the ADER framework for MARL to handle multi-agent exploration-exploitation trade-off. The proposed method is based on entropy regularization with learning proper target entropy values across agents over time by using a newly-proposed metric to measure the relative benefit of more exploration for each agent. Numerical results on various tasks including the sparse SMAC environment show that ADER can properly handle time-varying multi-agent exploration-exploitation trade-off effectively and outperforms other state-of-the-art baselines. Furthermore, we expect the key ideas of ADER can be applied to other exploration methods for MARL such as intrinsic motivation.

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

## A  APPENDIX A: PROOFS

**Proposition 2** *The decomposed soft Bellman operators $\mathcal{T}_R^\pi$ and $\mathcal{T}_{H,i}^\pi$ are contractions.*

*Proof:*  The action value functions $Q_{JT}^R(\boldsymbol{\tau_t}, \boldsymbol{a_t})$ and $Q_{JT}^{H,i}(\boldsymbol{\tau_t}, \boldsymbol{a_t})$ can be estimated based on their corresponding Bellman backup operators, defined by

$$\mathcal{T}_R^\pi Q_{JT}^R(s_t, \boldsymbol{\tau_t}, \boldsymbol{a_t}) := r_t + \gamma \mathbb{E}\left[V_{JT}^R(s_{t+1}, \boldsymbol{\tau_{t+1}})\right], \quad \text{where} \tag{18}$$
$$V_{JT}^R(s_t, \boldsymbol{\tau_t}) = \mathbb{E}\left[Q_{JT}^R(s_t, \boldsymbol{\tau_t}, \boldsymbol{a_t})\right]$$
$$\mathcal{T}_{H,i}^\pi Q_{JT}^{H,i}(s_t, \boldsymbol{\tau_t}, \boldsymbol{a_t}) := \gamma \mathbb{E}\left[V_{JT}^{H,i}(s_{t+1}, \boldsymbol{\tau_{t+1}})\right], \quad \text{where} \tag{19}$$
$$V_{JT}^{H,i}(s_t, \boldsymbol{\tau_t}) = \mathbb{E}\left[Q_{JT}^{H,i}(s_t, \boldsymbol{\tau_t}, \boldsymbol{a_t}) - \alpha^i \log \pi(a_t^i | \tau_t^i)\right].$$

Here, $V_{JT}^R(s_t, \boldsymbol{\tau_t})$ and $V_{JT}^{H,i}(s_t, \boldsymbol{\tau_t})$ are the joint value functions regarding reward and entropy, respectively.

First, let us consider the decomposed Bellman operator regarding reward, $\mathcal{T}_R^\pi$. For the sake of simplicity, we abbreviate $(Q_{JT}^R, Q_{JT}^{H,i}, V_{JT}^R, V_{JT}^{H,i})$ as $(Q^R, Q^{H,i}, V^R, V^{H,i})$. From (18), we have

$$\mathcal{T}_R^\pi Q^R(s_t, \boldsymbol{\tau_t}, \boldsymbol{a_t}) = r_t + \gamma \mathbb{E}_{s_{t+1}, \boldsymbol{\tau_{t+1}}, \boldsymbol{a_{t+1}}}\left[Q^R(s_{t+1}, \boldsymbol{\tau_{t+1}}, \boldsymbol{a_{t+1}})\right]. \tag{20}$$

Then, we have

$$\|\mathcal{T}_R^\pi(q_t^1) - \mathcal{T}_R^\pi(q_t^2)\|_\infty$$
$$= \|(r_t + \gamma \sum_{\substack{s_{t+1}, \boldsymbol{\tau_{t+1}} \\ \boldsymbol{a_{t+1}}}} \boldsymbol{\pi}(\boldsymbol{a_{t+1}}|\boldsymbol{\tau_{t+1}}) p(s_{t+1}, \boldsymbol{\tau_{t+1}}|s_t, \boldsymbol{\tau_t}, \boldsymbol{a_t}) \cdot q_{t+1}^1)$$
$$- (r_t + \gamma \sum_{\substack{s_{t+1}, \boldsymbol{\tau_{t+1}} \\ \boldsymbol{a_{t+1}}}} \boldsymbol{\pi}(\boldsymbol{a_{t+1}}|\boldsymbol{\tau_{t+1}}) p(s_{t+1}, \boldsymbol{\tau_{t+1}}|s_t, \boldsymbol{\tau_t}, \boldsymbol{a_t}) \cdot q_{t+1}^2)\|_\infty$$
$$= \|\gamma \sum_{\substack{s_{t+1}, \boldsymbol{\tau_{t+1}} \\ \boldsymbol{a_{t+1}}}} \boldsymbol{\pi}(\boldsymbol{a_{t+1}}|\boldsymbol{\tau_{t+1}}) p(s_{t+1}, \boldsymbol{\tau_{t+1}}|s_t, \boldsymbol{\tau_t}, \boldsymbol{a_t}) \cdot (q_{t+1}^1 - q_{t+1}^2))\|_\infty$$
$$\leq \|\gamma \sum_{\substack{s_{t+1}, \boldsymbol{\tau_{t+1}} \\ \boldsymbol{a_{t+1}}}} \boldsymbol{\pi}(\boldsymbol{a_{t+1}}|\boldsymbol{\tau_{t+1}}) p(s_{t+1}, \boldsymbol{\tau_{t+1}}|s_t, \boldsymbol{\tau_t}, \boldsymbol{a_t})\|_\infty \|q_{t+1}^1 - q_{t+1}^2\|_\infty$$
$$\leq \gamma \|q_{t+1}^1 - q_{t+1}^2\|_\infty$$

for $q_t^1 = \left[Q_1^R(s_t, \boldsymbol{\tau_t}, \boldsymbol{a_t})\right]_{\substack{s_t \in \mathcal{S}, \boldsymbol{a_t} \in \mathcal{A} \\ \boldsymbol{\tau_t} \in (\Omega \times \mathcal{A})^*}}$ and $q_t^2 = \left[Q_2^R(s_t, \boldsymbol{\tau_t}, \boldsymbol{a_t})\right]_{\substack{s_t \in \mathcal{S}, \boldsymbol{a_t} \in \mathcal{A} \\ \boldsymbol{\tau_t} \in (\Omega \times \mathcal{A})^*}}$ since $\|\sum_{\substack{s_{t+1}, \boldsymbol{\tau_{t+1}} \\ \boldsymbol{a_{t+1}}}} \boldsymbol{\pi}(\boldsymbol{a_{t+1}}|\boldsymbol{\tau_{t+1}}) p(s_{t+1}, \boldsymbol{\tau_{t+1}}|s_t, \boldsymbol{\tau_t}, \boldsymbol{a_t})\|_\infty \leq 1$. Thus, the operator $\mathcal{T}_R^\pi$ is a $\gamma$-contraction.

Next, let us consider the decomposed Bellman operator regarding entropy, $\mathcal{T}_{H,i}^\pi$. From (19), we have

$$\mathcal{T}_{H,i}^\pi Q^{H,i}(s_t, \boldsymbol{\tau_t}, \boldsymbol{a_t}) = \gamma \mathbb{E}\left[Q^{H,i}(s_{t+1}, \boldsymbol{\tau_{t+1}}, \boldsymbol{a_{t+1}}) - \alpha^i \log \pi(a_{t+1}^i | \tau_{t+1}^i))\right]. \tag{21}$$

Then, we have

$$\|\mathcal{T}_{H,i}^{\pi}(q_t^1) - \mathcal{T}_{H,i}^{\pi}(q_t^2)\|_{\infty}$$

$$= \|(\gamma \sum_{\substack{s_{t+1}, \boldsymbol{\tau_{t+1}} \\ \boldsymbol{a_{t+1}}}} \boldsymbol{\pi}(\boldsymbol{a_{t+1}}|\boldsymbol{\tau_{t+1}})p(s_{t+1}, \boldsymbol{\tau_{t+1}}|s_t, \boldsymbol{\tau_t}, \boldsymbol{a_t}) \cdot (q_{t+1}^1 - \alpha^i \log \pi(a_{t+1}^i|\tau_{t+1}^i))$$

$$- (\gamma \sum_{\substack{s_{t+1}, \boldsymbol{\tau_{t+1}} \\ \boldsymbol{a_{t+1}}}} \boldsymbol{\pi}(\boldsymbol{a_{t+1}}|\boldsymbol{\tau_{t+1}})p(s_{t+1}, \boldsymbol{\tau_{t+1}}|s_t, \boldsymbol{\tau_t}, \boldsymbol{a_t}) \cdot (q_{t+1}^2 - \alpha^i \log \pi(a_{t+1}^i|\tau_{t+1}^i))\|_{\infty}$$

$$= \|\gamma \sum_{\substack{s_{t+1}, \boldsymbol{\tau_{t+1}} \\ \boldsymbol{a_{t+1}}}} \boldsymbol{\pi}(\boldsymbol{a_{t+1}}|\boldsymbol{\tau_{t+1}})p(s_{t+1}, \boldsymbol{\tau_{t+1}}|s_t, \boldsymbol{\tau_t}, \boldsymbol{a_t}) \cdot (q_{t+1}^1 - q_{t+1}^2))\|_{\infty}$$

$$\leq \|\gamma \sum_{\substack{s_{t+1}, \boldsymbol{\tau_{t+1}} \\ \boldsymbol{a_{t+1}}}} \boldsymbol{\pi}(\boldsymbol{a_{t+1}}|\boldsymbol{\tau_{t+1}})p(s_{t+1}, \boldsymbol{\tau_{t+1}}|s_t, \boldsymbol{\tau_t}, \boldsymbol{a_t})\|_{\infty} \|q_{t+1}^1 - q_{t+1}^2\|_{\infty}$$

$$\leq \gamma \|q_{t+1}^1 - q_{t+1}^2\|_{\infty}$$

for $q_t^1 = \left[Q_1^R(s_t, \boldsymbol{\tau_t}, \boldsymbol{a_t})\right]_{\substack{s_t \in \mathcal{S}, \boldsymbol{a_t} \in \mathcal{A} \\ \boldsymbol{\tau_t} \in (\Omega \times \mathcal{A})^*}}$ and $q_t^2 = \left[Q_2^R(s_t, \boldsymbol{\tau_t}, \boldsymbol{a_t})\right]_{\substack{s_t \in \mathcal{S}, \boldsymbol{a_t} \in \mathcal{A} \\ \boldsymbol{\tau_t} \in (\Omega \times \mathcal{A})^*}}$ since $\|\sum_{\substack{s_{t+1}, \boldsymbol{\tau_{t+1}} \\ \boldsymbol{a_{t+1}}}} \boldsymbol{\pi}(\boldsymbol{a_{t+1}}|\boldsymbol{\tau_{t+1}})p(s_{t+1}, \boldsymbol{\tau_{t+1}}|s_t, \boldsymbol{\tau_t}, \boldsymbol{a_t})\|_{\infty} \leq 1$. Thus, the operator $\mathcal{T}_{H,i}^{\pi}$ is a $\gamma$-contraction.

## B   APPENDIX B: DETAILED IMPLEMENTATION

Here, we describe the implementation of ADER for discrete action tasks based on SAC-discrete (Christodoulou, 2019). The learning process consists of the update of both temperature parameters and target entropies and the approximation of multi-agent maximum entropy solution, which consists of the update of the joint policy and the critics. To do this, we first approximate the policies $\{\pi^i_{\phi_i}\}^N_{i=1}$, the joint action value functions $Q^R_{JT,\theta_R}$ and $Q^{H,i}_{JT,\theta_{H,i}}$ by using deep neural networks with parameters, $\{\phi_i\}^N_{i=1}$, $\theta_R$ and $\{\theta_{H,i}\}^N_{i=1}$.

First, the joint policy is updated based on Eq. (12) and the loss function is given by

$$L(\boldsymbol{\phi}) = \mathbb{E}_{(s_t,\boldsymbol{\tau_t})\sim\mathcal{D},\{a^i_t\sim\pi^i(\cdot|\tau^i_t)\}^N_{i=1}}\Bigg[\sum^N_{i=1}\alpha^i(\log\pi^i_{\phi_i}(a^i_t|\tau^i_t) - Q^{H,i}_{JT,\theta_{H,i}}(s_t,\boldsymbol{\tau_t},\boldsymbol{a_t}))$$
$$- Q^R_{JT,\theta_R}(s_t,\boldsymbol{\tau_t},\boldsymbol{a_t})\Bigg], \qquad \text{(B.1)}$$

where $\boldsymbol{\phi} = \{\phi_i\}^N_{i=1}$ is the parameter for the joint policy. Next, the joint action value functions are trained based on the disentangled Bellman operators defined in Eq. (10) and the loss functions are given by

$$L(\theta_R) = \mathbb{E}_{(s_t,\boldsymbol{\tau_t},\boldsymbol{a_t},s_{t+1},\boldsymbol{\tau_{t+1}})\sim\mathcal{D}}\Bigg[\frac{1}{2}(Q^R_{JT,\theta_R}(s_t,\boldsymbol{\tau_t},\boldsymbol{a_t}) - (r_t + \gamma V^R_{JT,\bar{\theta}_R}(s_{t+1},\boldsymbol{\tau_{t+1}})))^2\Bigg]$$
$$\text{(B.2)}$$

$$L(\theta_{H,i}) = \mathbb{E}_{(s_t,\boldsymbol{\tau_t},\boldsymbol{a_t},s_{t+1},\boldsymbol{\tau_{t+1}})\sim\mathcal{D}}\Bigg[\frac{1}{2}(Q^{H,i}_{JT,\theta_i}(s_t,\boldsymbol{\tau_t},\boldsymbol{a_t}) - \gamma V^{H,i}_{JT,\bar{\theta}_{H,i}}(s_{t+1},\boldsymbol{\tau_{t+1}})))^2\Bigg] \qquad \text{(B.3)}$$

where $V^R_{JT,\bar{\theta}_R}$ and $V^{H,i}_{JT,\bar{\theta}_{H,i}}$ are defined as follows:

$$V^R_{JT,\bar{\theta}_R}(s_t,\boldsymbol{\tau_t}) = \mathbb{E}\left[Q^R_{JT,\bar{\theta}_R}(s_t,\boldsymbol{\tau_t},\boldsymbol{a_t})\right] \qquad \text{(B.4)}$$

$$V^{H,i}_{JT,\bar{\theta}_{H,i}}(s_t,\boldsymbol{\tau_t}) = \mathbb{E}\left[Q^{H,i}_{JT,\bar{\theta}_{H,i}}(s_t,\boldsymbol{\tau_t},\boldsymbol{a_t}) - \alpha^i\log\pi(a^i_t|\tau^i_t)\right]. \qquad \text{(B.5)}$$

Note that $\bar{\theta}_R$ and $\bar{\theta}_{H,i}$ are obtained based on the EMA of the parameters of the joint action-value functions. Although the definitions of the state value functions are given by (B.4) and (B.5), we do not use this definition to compute the state value functions. This is because the marginalization over joint action becomes complex as the number of agents increases. For the practical computation of $V^R_{JT}$ and $V^{H,i}_{JT}$, we do the following for reduced complexity. We first marginalize the individual $Q$-function based on individual action to get $V^R_i$. Then, we feed $V^R_1,\cdots,V^R_N$ of all agents to the mixing network $f^{V,R}_{mix}$ to obtain the joint state value function as $V^R_{JT}(s,\boldsymbol{\tau}) = f^{V,R}_{mix}(s,V^R_1(\tau^1),\cdots,V^R_N(\tau^N))$. Here, $f^{V,R}_{mix}$ is learned such that $f^{V,R}_{mix}$ follows the definition by the TD loss eq. (B.2) and the Bellman equation. In addition, we share the mixing network for $Q^{H,i}_{JT}$ for all $i\in\mathcal{N}$ and inject the one-hot vector which denotes the agent index $i$ to handle the scalability.

We update the temperature parameters based on Eq. (13) and the loss function is given by

$$L(\alpha^i) = \mathbb{E}_{\boldsymbol{\tau_t}\sim\mathcal{D},\{a^i_t\sim\pi^i(\cdot|\tau^i_t)\}^N_{i=1}}\left[-\alpha^i\log\boldsymbol{\pi}_t(a^i_t|\tau^i_t) - \alpha^i\mathcal{H}_i\right], \quad \forall i\in\mathcal{N}. \qquad \text{(B.6)}$$

Finally, we update the target entropy of each agent. For $\mathcal{H}_0\geq 0$, we set the coefficients $\beta_i$ for determining the individual target entropy $\mathcal{H}_i$ as $\boldsymbol{\beta} = \left[\beta_1,\cdots,\beta_i,\cdots,\beta_N\right] =$

$$\text{Softmax}\left[\mathbb{E}\left[\frac{\partial V^R_{JT}(s,\boldsymbol{\tau})}{\partial\mathcal{H}(\pi^1_t(\cdot|\tau^1))}\right],\cdots,\mathbb{E}\left[\frac{\partial V^R_{JT}(s,\boldsymbol{\tau})}{\partial\mathcal{H}(\pi^i_t(\cdot|\tau^i))}\right],\cdots,\mathbb{E}\left[\frac{\partial V^R_{JT}(s,\boldsymbol{\tau})}{\partial\mathcal{H}(\pi^N_t(\cdot|\tau^N))}\right]\right], \qquad \text{(B.7)}$$

where the computation of $\frac{\partial V^R_{JT}(s,\tau)}{\partial\mathcal{H}(\pi^i_t(\cdot|\tau^i))}$ is explained in Section B.1.

Note that we change the sign of the elements in Eq. (B.7) if $\mathcal{H}_0 < 0$ to satisfy the core idea of ADER, which assigns a high target entropy to the agent whose benefit to the joint value is small.

In addition, before the softmax layer, we normalize the elements in Eq. (B.7). Based on the coefficients, the target entropy is given by $\mathcal{H}_i = \beta_i^{EMA} \times \mathcal{H}_0$ where $\beta_i^{EMA}$ is computed recursively as

$$\boldsymbol{\beta}^{EMA} \leftarrow (1-\xi)\boldsymbol{\beta}^{EMA} + \xi\boldsymbol{\beta} \tag{B.8}$$

## B.1 Computation of the metric $\frac{\partial V_{JT}^R(s,\tau)}{\partial \mathcal{H}(\pi_t^i(\cdot|\tau^i))}$

We adopted an actor-critic structure for our algorithm. Hence, for each agent we have a separate actor, i.e., policy in both continuous-action and discrete-action cases, as seen in Figures 6 and 7, which show the overall structure for continuous-action and discrete-action cases, respectively. The computation of the partial derivative $\frac{\partial V_{JT}^R(s,\tau)}{\partial \mathcal{H}(\pi_t^i(\cdot|\tau^i))}$ in Eq. (B.7) depends on the overall structure, especially on the structure of the individual critic network.

First, consider the continuous-action case. In this case, we used a Gaussian policy for each agent. Then, the policy neural network of Agent $i$ with trainable parameter $\theta^i$ takes trajectory $\tau_t^i$ as input and generates the mean $\mu^i$ and the log variance $\log \sigma^i$ as output, as shown in Figure 6. Based on these outputs and the reparameterization trick, the action of Agent $i$ is generated as $a^i = \mu^i + \exp(\log \sigma^i)Z^i$, where $Z^i$ is Gaussian-distributed with zero mean and identity covariance matrix, i.e., $Z^i \sim N(0, I)$. The action $a^i$ and trajectory $\tau^i$ are applied as input to both return and entropy critic networks for Agent $i$, as seen in Figure 6. Now, focus on the return critic network of Agent $i$, which is relevant to the computation of our metric. The return critic of Agent $i$ generates the local Q-value $Q_i^R(\tau^i, a^i)$. All local $Q$-values $Q_1^R(\tau^1, a^1), \cdots, Q_N^R(\tau^N, a^N)$ from all agents are applied as input to the mixing network for global return value $Q_{JT}^R$, as seen in Figure 6. Due to the connected tensor structure in Figure 6, at the time of learning, the gradient of $Q_{JT}^R$ with respect to $\log \sigma^i$ can be computed by deep learning libraries such as Pytorch. Note that $\log \sigma^i$ is simply a scaled version of the Gaussian policy entropy. So, we can just obtain this value $\partial Q_{JT}^R / \partial \log \sigma^i$ from deep learning libraries. Furthermore, $V_{JT}^R$ can be obtained by sampling multiple $a^i$'s from the same policy $\pi_i^i$, computing the corresponding multiple $Q$-values and taking the average over the multiple $a^i$ samples. However, we simplify this step and just use $\partial Q_{JT}^R / \partial \log \sigma^i$ as our estimate for the metric $\frac{\partial V_{JT}^R(s,\tau)}{\partial \mathcal{H}(\pi_t^i(\cdot|\tau^i))}$. Indeed, many algorithms use single-sample average for obtaining expectations for algorithm simplicity.

Second, consider the discrete-action case. In this case, we again use an actor-critic structure for our algorithm. The structure of the critic network of Agent $i$ in the discrete-action case is different from that in the continuous-action case. Whereas the critic network takes the trajectory $\tau^i$ and the action $a^i$ as input, and generates $Q_i^R(\tau^i, a^i)$ in the continuous-action case, the critic network typically uses the DQN structure (Mnih et al. 2015), which takes the trajectory $\tau^i$ as input and generates all $Q_i^R(\tau^i, a_1^i), \cdots, Q_i^R(\tau^i, a_{|\mathcal{A}|}^i)$ as output in the discrete-action case. In the discrete-action case, action is over a finite action set $\mathcal{A} = \{a_1, \cdots, a_{|\mathcal{A}|}\}$, and the policy is described by a categorical distribution $\mathbf{p}^i = \left[p_1^i, \cdots, p_{|\mathcal{A}|}^i\right]$ over $\mathcal{A}$ for each state (or trajectory). Hence, our actor, i.e, policy $\pi^i$ for Agent $i$ is a deep neural network which takes the observation $\tau^i$ as input and generates probability vector $\mathbf{p}^i = \left[p_1^i, \cdots, p_{|\mathcal{A}|}^i\right]$ as output. Here, let us denote the policy deep neural network parameter by $\theta^i$ and denote the policy $\pi_t^i$ by $\pi_{\theta^i}^i$, showing the current parameter explicitly. Then, using the output $\mathbf{p}^i = \left[p_1^i, \cdots, p_{|\mathcal{A}|}^i\right]$ of the policy network and the output $Q_i^R(\tau^i, a_1^i), \cdots, Q_i^R(\tau^i, a_{|\mathcal{A}|}^i)$ of the critic network, we compute the local return value as

$$V_i^R(\tau^i) = \sum_{j=1}^{|\mathcal{A}|} p_j^i(\tau^i) Q_i^R(\tau^i, a_j^i). \tag{22}$$

Then, all local return values $V_1^R(\tau^1), \cdots, V_N^R(\tau^N)$ are fed to the mixing network for global return value $V_{JT}^R$, as seen in Figure 7.

In this discrete-action case, the policy entropy is given by $\mathcal{H}(\pi_{\theta^i}^i(\cdot|\tau^i)) = -\sum_{j=1}^{N} p_j^i \log p_j^i$. On the contrary to the continuous-action case in which the policy entropy $\log \sigma^i$ is an explicit node value in the overall structure and hence the output $V_{JT}^R$ gradient with respect to the node $\log \sigma^i$ is directly available, in the discrete-action case there is no node corresponding to the value $\mathcal{H}(\pi_{\theta^i}^i(\cdot|\tau^i)) = -\sum_{j=1}^{N} p_j^i \log p_j^i$. Hence, the gradient $\frac{\partial V_{JT}^R}{\partial \mathcal{H}(\pi_{\theta^i}^i)}$ is not readily available from the architecture. Note that we only have nodes for $p_1^i, \cdots, p_{|\mathcal{A}|}^i$ in the architecture, but the gradient of $V_{JT}^R$ with respect to $p_j^i$ is not $\frac{\partial V_{JT}^R}{\partial \mathcal{H}(\pi_{\theta^i}^i)}$. Furthermore, it is not easy to compute $\frac{\partial V_{JT}^R}{\partial \mathcal{H}(\pi_{\theta^i}^i)}$ from $\frac{\partial V_{JT}^R}{\partial p_j^i}$, $j = 1, \cdots, |\mathcal{A}|$ with $\sum_j p_j^i = 1$ for general cardinality $|\mathcal{A}|$.

To circumvent this difficulty and compute the metric $\frac{\partial V_{JT}^R}{\partial \mathcal{H}(\pi_{\theta^i}^i)}$, we exploit the policy network parameter $\theta^i$ and numerical computation. When the current policy network parameter is $\theta^i$, we have the corresponding policy network output $p_1^i, \cdots, p_{|\mathcal{A}|}^i$. Then, consider the temporary scalar objective function $\mathcal{H}(\pi_{\theta^i}^i)$ for the policy network. We can compute the gradient of $\mathcal{H}(\pi_{\theta^i}^i)$ with respect to the policy parameter $\theta^i$. Let us denote this gradient by $\frac{\partial \mathcal{H}(\pi_{\theta^i}^i)}{\partial \theta^i}$, which is the direction of $\theta^i$ for maximum policy entropy increase. Then, we update the policy parameter as $\tilde{\theta}^i = \theta^i + \delta \frac{\partial \mathcal{H}(\pi_{\theta^i}^i)}{\partial \theta^i}$, where $\delta$ is a positive stepsize. Then, for the updated policy $\pi_{\tilde{\theta}^i}^i$, we compute the corresponding $p_1^i, \cdots, p_{|\mathcal{A}|}^i$. Using these updated probability values, we compute the local value $V_i^R$ by using eq. (22). Using the values before and after the update, we compute $\frac{\Delta V_i^R(\tau^i)}{\Delta \mathcal{H}(\pi^i)} = \frac{V_i^R(\tau^i; \pi_{\tilde{\theta}^i}^i) - V_i^R(\tau^i; \pi_{\theta^i}^i)}{\mathcal{H}(\pi_{\tilde{\theta}^i}^i) - \mathcal{H}(\pi_{\theta^i}^i)}$.

Now, the metric $\frac{\partial V_{JT}^R}{\partial \mathcal{H}(\pi_{\theta^i}^i)}$ can be computed based on the chain rule. That is, we have $\frac{\partial V_{JT}^R}{\partial \mathcal{H}(\pi_{\theta^i}^i)} = \frac{\partial V_{JT}^R(s,\boldsymbol{\tau})}{\partial V_i^R(\tau^i)} \times \frac{\partial V_i^R(\tau^i)}{\partial \mathcal{H}(\pi_{\theta^i}^i)}$. Here, the first term $\frac{\partial V_{JT}^R(s,\boldsymbol{\tau})}{\partial V_i^R(\tau^i)}$ is available from deep learning libraries since $V_{JT}^R$ and $V_i^R$ are nodes of the learning architecture. The second term $\frac{\partial V_i^R(\tau^i)}{\partial \mathcal{H}(\pi_{\theta^i}^i)}$ can be approximated by $\frac{\Delta V_i^R(\tau^i)}{\Delta \mathcal{H}(\pi^i)}$ in the above.

Note that the policy update $\tilde{\theta}^i = \theta^i + \delta \frac{\partial \mathcal{H}(\pi_{\theta^i}^i)}{\partial \theta^i}$ is only for computation of the metric. It is not done for the actual learning update.

## B.2 Overall Architecture and Algorithm Pseudocode

We summarize the proposed algorithm in Algorithm 1 and illustrate the overall architecture of the proposed ADER in Figures 6 and 7.

---

**Algorithm 1** **AD**aptive **E**ntropy-**R**egularization for multi-agent reinforcement learning (ADER)

Initialize parameters $\{\phi_i\}_{i=1}^N$, $\theta_R$, $\{\theta_{H,i}\}_{i=1}^N$, $\bar{\theta}_R$, $\{\bar{\theta}_{H,i}\}_{i=1}^N$
Generate a trajectory $\tau$ by interacting with the environment by using the joint policy $\boldsymbol{\pi}$ and store $\tau$ in the replay memory
**for** $episode = 1, 2, \cdots$ **do**
    Generate a trajectory $\tau$ by using the joint policy $\boldsymbol{\pi}$ and store $\tau$ in the replay memory $\mathcal{D}$
    **for** each gradient step **do**
        Sample a minibatch from $\mathcal{D}$
        Update $\{\phi_i\}_{i=1}^N$ by minimizing the loss function Eq. (B.1)
        Update $\theta_R$, $\{\theta_{H,i}\}_{i=1}^N$ by minimizing the loss functions Eq. (B.2) and Eq. (B.3)
        Update $\alpha^i$ by minimizing the loss function Eq. (B.6)
        Update $\{\mathcal{H}_i\}_{i=1}^N$ by computing Eq. (B.7) and Eq. (B.8)
        Update $\bar{\theta}_R$ and $\{\bar{\theta}_{H,i}\}_{i=1}^N$ by EMA based on $\theta_R$ and $\{\theta_{H,i}\}_{i=1}^N$
    **end for**
**end for**

---

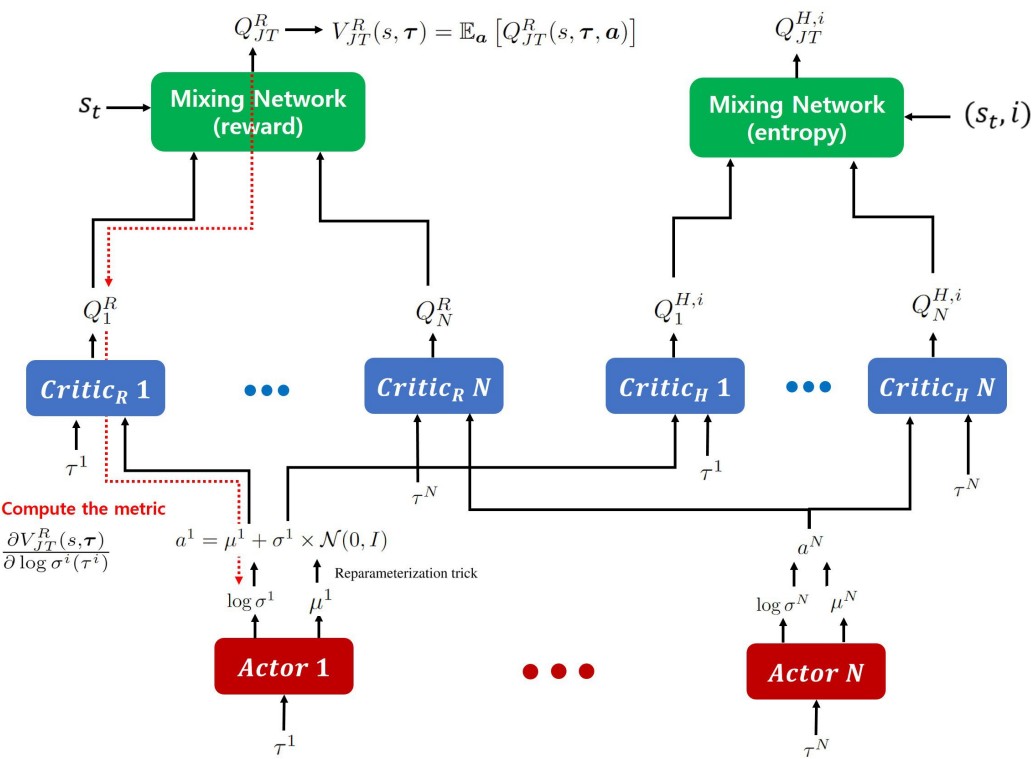

Figure 6: Overall architecture of ADER in continuous action cases

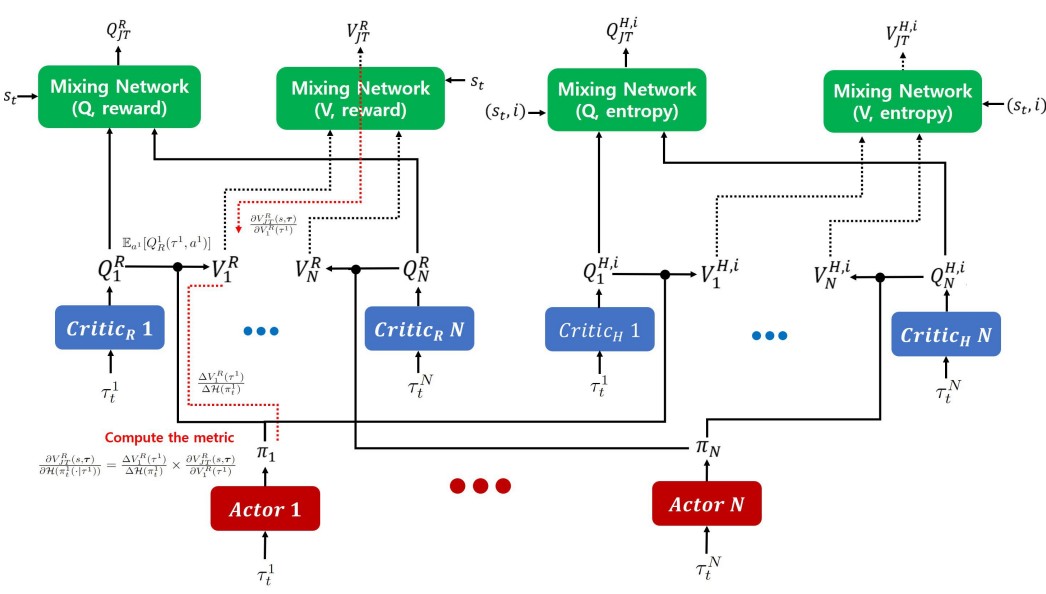

Figure 7: Overall architecture of ADER in discrete action cases

APPENDIX C: TRAINING DETAILS

We compute the joint value function as $V_{JT}^R(s, \tau) = f_{mix}^{V,R}(s, V_1^R(\tau^1), \cdots, V_N^R(\tau^N))$. To compute this, as similar in (Zhang et al., 2021), we first obtain the local value functions as $V_i^R(\tau^i) = \mathbb{E}_{a^i}[Q^R(\tau^i, a^i)]$ and then input the obtained local value functions into the mixing network. For discrete action environments, we share the mixing network for both $V_{JT}^R$ and $Q_{JT}^R$, and thus the mixing network is trained to minimize the TD error of $Q_{JT}^R$. It works well as the reviewer can see in the experimental results. For continuous action environments, we use two mixing networks for $V_{JT}^R$ and $Q_{JT}^R$ which are trained separately as in SAC (Haarnoja et al., 2018a). In addition, we need N mixing networks for $Q_{JT}^{H,i}$. To handle the scalability, we share the mixing network for $Q_{JT}^{H,i}$ for all $i \in \mathcal{N}$ and inject the one-hot vector which denotes the agent index $i$ as QMIX shares the local Q-functions with one parameterized neural network.

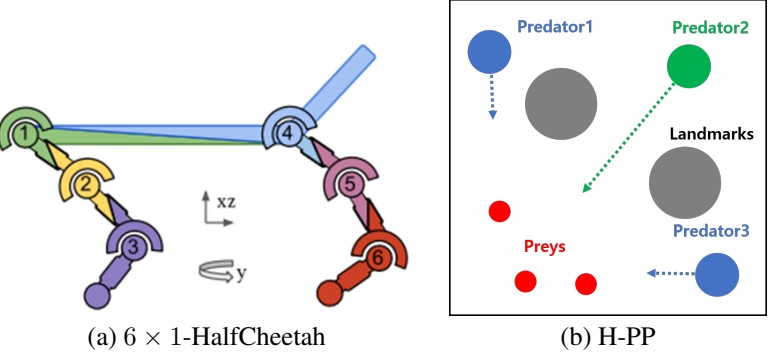

(a) $6 \times 1$-HalfCheetah      (b) H-PP

Figure 8: Considered continuous action tasks

## C1. Environment Details

**Multi-agent HalfCheetah**  We considered the multi-agent HalfCheetah introduced in (Peng et al., 2021). As illustrated in Fig. 8 (a), the multi-agent HalfCheetah divides the body into disjoint sub-graphs and each sub-graph corresponds to an agent. We used $6 \times 1$-HalfCheetah, which consists of six agents with one action dimension. We set the maximum graph distance $k = 1$, where $k$ denotes the distance each agent can observe. We set the maximum episode length as $T_{max} = 1000$.

**Heterogeneous Predator-Prey (H-PP)**  We modified the continuous predator-prey environment considered in (Peng et al., 2021) to be heterogeneous. As illustrated in Fig. 8 (b), the considered heterogeneous predator-prey consists of three predator agents, where the maximum speeds of an agent ($v_{max}^1 = 1.0$) and other agents ($v_{max}^2 = 0.75$) are different, three preys with the maximum speed ($v_{max}^3 = 1.25$) is faster than all predators and the landmarks. The preys move away from the nearest predator implemented in (Peng et al., 2021) and thus the predators should be trained to pick one prey and catch the prey together. Each agent observes the relative positions of the other predators and the landmarks within view range and the relative positions and velocities of the prey within view range. The reward $+10$ is given when one of the predators collides with the prey. We set the maximum episode length as $T_{max} = 50$.

**Starcraft II**  We evaluated ADER on the StarcraftII micromanagement benchmark (SMAC) environment (Samvelyan et al., 2019). To make the problem more difficult, we modified the SMAC environment to be sparse. The considered sparse reward setting consists of a death reward and time-penalty reward. The time-penalty reward is $-0.1$ and the death reward is given $+10$ and $-1$ when one enemy dies and one ally dies, respectively. Additionally, the dead reward is given $+200$ if all enemies die.

## C2. Training Details and Hyperparameters

We implemented ADER based on (Samvelyan et al., 2019; Peng et al., 2021; Zhang et al., 2021) and conducted the experiments on a server with Intel(R) Xeon(R) Gold 6240R CPU @ 2.40GHz and 8

Nvidia Titan xp GPUs. Each experiment took about 12 to 24 hours. We used the implementations of the considered baselines provided by the authors.

**Multi-agent HalfCheetah**    In the multi-agent halfcheetah environment, the architecture of the policies and critics for ADER follows (Peng et al., 2021). We use an MLP with 2 hidden layers which have 400 and 300 hidden units and ReLU activation functions. The final layer uses tanh activation function to bound the action as in (Haarnoja et al., 2018a). We also use the reparameterization trick for the policy as in (Haarnoja et al., 2018a). The replay buffer stores up to $10^6$ transitions and 100 transitions are uniformly sampled for training. As in (Haarnoja et al., 2018b), we set the sum of target entropy as

$$\mathcal{H}_0 = N \times (-dim(\mathcal{A})) = 6 \times (-1) = -6,$$

where $N$ is the number of agents. We set the hyperparameter for EMA filter as $\xi = 0.9$ and initialize the temperature parameters as $\alpha_{init}^i = e^{-2}$ for all $i \in \mathcal{N}$.

**Heterogeneous Predator-Prey**    In the heterogeneous predator-prey environment, the architecture of the policies and critics for ADER follows (Peng et al., 2021). To parameterize the policy, we use a deep neural network which consists of a fully-connected layer, GRU and a fully-connected layer which have 64 dimensional hidden units. The final layer uses tanh activation function to bound the action. Next, for the critic network, we use a MLP with 2 hidden layers which have 64 hidden units and ReLU activation function. The replay buffer stores up to 5000 episodes and 32 episodes are uniformly sampled for training. As in (Haarnoja et al., 2018b), we set the sum of target entropy as

$$\mathcal{H}_0 = N \times (-dim(\mathcal{A})) = 3 \times (-2) = -6.$$

We set the hyperparameter for EMA filter as $\xi = 0.9$ and initialize the temperature parameters as $\alpha_{init}^i = e^{-2}$ for all $i \in \mathcal{N}$.

**Starcraft II**    For parameterization of the policy we use a deep neural network which consists of a fully-connected layer, GRU and a fully-connected layer which have 64 dimensional hidden units. For the critic networks we use a MLP with 2 hidden layers which have 64 hidden units and ReLU activation function. The replay buffer stores up to 5000 episodes and 32 episodes are uniformly sampled for training. For the considered maps in SMAC, we use different hyperparameters. We set the sum of target entropy based on the maximum entropy, which can be achieved if the policy is uniform distribution, as

$$\mathcal{H}_0 = N \times \mathcal{H}^* \times k_{ratio} = N \times \log(dim(\mathcal{A})) \times k_{ratio}.$$

The values of $k_{ratio}$, $\xi$, and initial temperature parameter for each map are summarized Table 1.

Table 1: Hyperparameters for the considered SMAC environment

| MAP | $k_{ratio}$ | $\xi$ | $\alpha_{init}^i$ |
|---|---|---|---|
| *1c3s5z* | 0.05 | 0.9 | $e^{-3}$ |
| *3m* | 0.1 | 0.9 | $e^{-2}$ |
| *3s5z* | 0.05 | 0.9 | $e^{-3}$ |
| *3s vs 3z* | 0.1 | 0.9 | $e^{-3}$ |
| *MMM2* | 0.1 | 0.9 | $e^{-2.5}$ |
| *8m vs 9m* | 0.1 | 0.9 | $e^{-3}$ |

In all the considered environments, we apply the value factorization technique proposed in (Rashid et al., 2018). The architecture of the mixing network for ADER, which follows (Rashid et al., 2018), takes the output of individual critics as input and outputs the joint action value function. The weights of the mixing network are produced by the hypernetwork which takes the global state as input. The hypernetwork consists of a MLP with a single hidden layer and an ELU activation function. Due to the ELU activation function, the weights of the mixing network are non-negative and this achieves the monotonic constraint in (Rashid et al., 2018). We expect that ADER can use other value factorization technique to yield better performance.

APPENDIX D: FURTHER EXPERIMENTS

**Experiments on the original SMAC environments**

We here provide the experiments on the original SMAC environments. We compared ADER with three baselines including FACMAC (Peng et al., 2021), FOP (Zhang et al., 2021) and QMIX (Rashid et al., 2018). For all the considered maps, ADER outperforms the baselines, as shown in Fig. 9. Thus, the proposed adaptive entropy-regularization method performs well in both original and sparse SMAC environments.

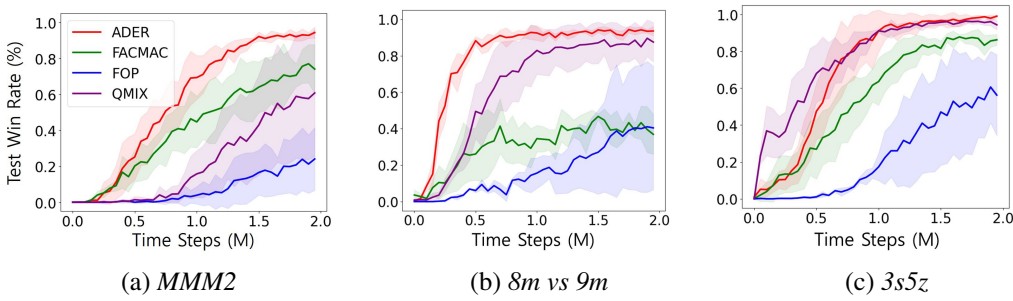

|     |     |     |
| (a) *MMM2* | (b) *8m vs 9m* | (c) *3s5z* |

Figure 9: Average test win rate on the original SMAC maps.

**Experiments on google research football environment**

We evaluated ADER on the google research football (GRF) environment, which is known as hard exploration tasks. We consider one scenario in GRF named *Academy 3 vs 1 with keeper*. In this environment, the agents receive a reward only when they succeed in scoring, which requires hard exploration. Thus, it is difficult to obtain the reward if all agents focus on exploration simultaneously.

We compared ADER with four baselines: QMIX, FOP, FACMAC, and SER-MARL. Fig. 10 shows the performance of ADER and the baselines, and the y-axis in Fig. B.2 denotes the median winning rate over 7 random seed. It is seen in Fig. 10 that ADER outperforms the baselines significantly. Since ADER handles multi-agent exploration-exploitation trade-off across multiple agents and over time, ADER performs better than SER-MARL, which keeps the same level of exploration across agents.

**Experiments on the modified SMAC environments**

Fig. 11 shows the performance of ADER and the considered seven baselines on the modified SMAC environment. It is seen that ADER outperforms all the considered baselines. Especially, on the hard tasks shown in Fig. 11, ADER significantly outperforms other baselines in terms of training speed and final performance. This is because those hard maps require high-quality adaptive exploration across agents over time. In the maps *3s vs 3z*, the stalkers (ally) should attack a zealot (enemy) many times and thus the considered reward is rarely obtained. In addition, since the stalker is a ranged attacker whereas the zealot is a melee attacker, the stalker should be trained to attack the zealot at a distance while avoiding the zealot. For this reason, if all stalkers focus on exploration simultaneously, they hardly remove the zealot, which leads to failure in solving the task. Similarly, in the hard tasks with imbalance between allies and enemies such as *MMM2*, and *8m vs 9m*, it is difficult to obtain a reward due to the simultaneous exploration of multiple agents. Thus, consideration of multi-agent exploration-exploitation trade-off is required to solve the task, and it seems that ADER effectively achieves this goal.

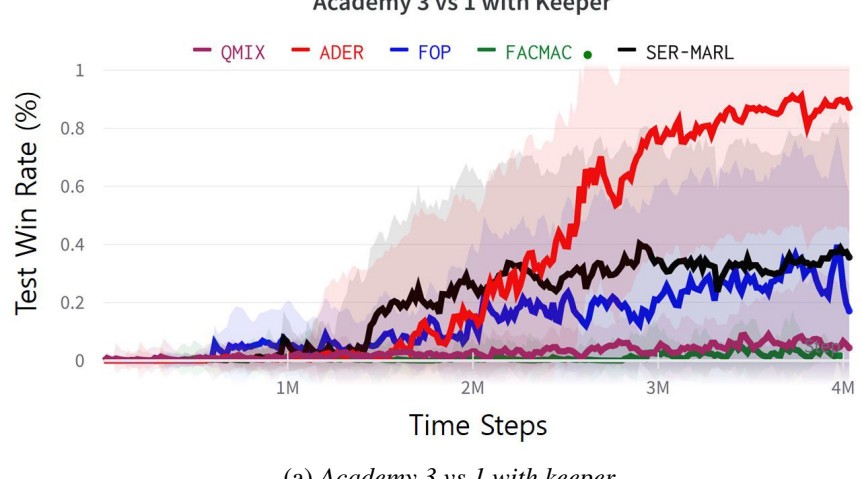

(a) *Academy 3 vs 1 with keeper*

Figure 10: Median test winning rate on Academy 3 vs 1 with keeper

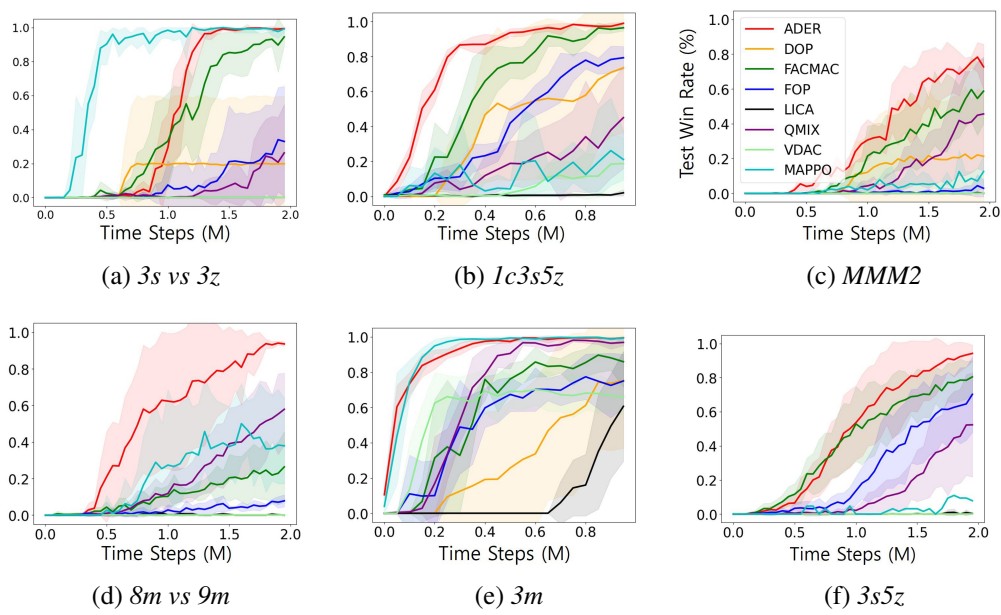

Figure 11: Average test win rate on the sparse SMAC maps.

## APPENDIX E: FURTHER RELATED WORKS

For effective exploration in single-agent RL, several approaches such as maximum entropy/entropy regularization (Haarnoja et al., 2017; 2018a), intrinsic motivation (Chentanez et al., 2004; Badia et al., 2019; Burda et al., 2018), parameter noise (Plappert et al., 2018; Fortunato et al., 2018) and count-based exploration (Ostrovski et al., 2017; Bellemare et al., 2016) have been considered. Also in MARL, exploration has been actively studied in various ways. MAVEN introduced a latent variable and maximized the mutual information between the latent variable and the trajectories to solve the poor exploration of QMIX caused by the representational constraint (Mahajan et al., 2019). Wang et al. (2019) proposes a coordinated exploration strategy by considering the interaction between agents. Liu et al. (2021b) proposes an efficient coordinated exploration method based

on restricted space selection to encourage multiple agents to explore worthy state space. Zheng et al. (2021) extends the intrinsic motivation-based exploration method to MARL and utilizes the episodic memory which stores highly rewarded episodes to boost learning. Gupta et al. (2021) promotes joint exploration by learning different tasks simultaneously based on multi-agent universal successor features to address the problem of relative overgeneralization. The aforementioned methods successfully improve exploration in MARL. However, to the best of our knowledge, none of the works address the multi-agent exploration-exploitation tradeoff, which is the main motivation of this paper.

## APPENDIX F: LIMITATION

In this paper, we only considered a fully cooperative setting where multiple agents share the global reward and showed that the proposed method successfully addresses the multi-agent exploration-exploitation tradeoff in such setting. However, the metric to measure the benefit of exploration can differ in other MARL settings such as mixed cooperative-competitive settings. Thus, we believe finding the metric in other MARL settings can be a good research direction.

