# OpenReview forum: "An Adaptive Entropy-Regularization Framework for Multi-Agent Reinforcement Learning"
_ICLR.cc/2023/Conference — Submitted to ICLR 2023_

### Official Review · Reviewer_mfnN · 2022-10-20

**Confidence:** 4
**Correctness:** 2
**Technical Novelty And Significance:** 2
**Empirical Novelty And Significance:** 2
**Recommendation:** 3

**Clarity, Quality, Novelty And Reproducibility:**

The Clarity of this paper can definitely be improved. Some references to equations are misused. Also, the authors should shorten the length of the motivation section and spend more paragraphs explaining the purpose of each component, as this is a relatively complex framework. Moreover, I think figure 6 can be improved by associating the loss function in it to better explain how each component is trained.

The Novelty of this paper heavily lies in the method of computing target entropy for each agent. However, as mentioned above, the computation of those partial derivative terms has to be further explained in order to judge the importance of this work.


**Strength And Weaknesses:**

### Strengths

1. This paper studies adaptive entropy regularization, which is important in entroy-regularized MARL and well-demonstrated in the motivation section.

2. The authors provide a comprehensive literature review of entropy-based MARL algorithms.

3. A variety of baselines is included in their SMAC experiments.

### Weaknesses

1. **Standard SMAC maps and more baselines should be included.** In the experiment section, MAPPO should also be included as a baseline, as it is one of the SOTA MARL algorithms. Also, experiments on the *standard* (unmodified) SMAC environment should be included, as it is a standard of all MARL papers. Moreover, for the matrix game, FOP should also be included as a baselines. As far as I understand, FOP uses a weight network to determine individual temperature parameters in their tasks except SMAC. That said, adaptive entropy regularization over agents and time is indeed considered in FOP.

2. **The writing can be largely improved.** Some paragraphs are too long, e.g., Section 1 is one paragraph, and Section 3.1 is one paragraph. Some references to equations are misused in section 3.4. The authors should refer to where those equations appear in the main text, not where they are in the appendix. For example, eq B.7 should be eq 15 in the main text.

4. **The presentation of the proposed method can be improved.** The overall workflow of this algorithm is complicated and unclear. An algorithm should be included to make it clear. Figure 6 does not provide a clear illustration.

### Questions

1. Two mixer networks are used in this method. The update of Q-mixer and $Q_i$ is common and easy to understand. However, It seems two methods are mentioned for computing $V_{jt}$. One is using V-mixer $V_{jt} = f_{mix}(s, V_1, \cdots V_n)$ and another is using expectation $V_{jt} = E[Q_{jt}]$. Do these two methods result in the same value? Or do you use those two values in different places?

2. As stated in eq 13, the policy update is achieved in a joint form using joint value function $Q_{jt}$, so the $Q_i$ will only be used for the computation of target entropy? If so, why do we need a mixer network for $Q_{jt}$, as we can directly parameterize $Q_{jt}$ via a single neural network? The learning of the proposed method is unclear in many ways, though I may misunderstand some parts.

3. As mentioned in Section 3.4, the partial derivative $\frac{\partial V_{jt}^{R}}{\partial H(\pi^i)}$ is used to determine the target entropy of agent $i$. For the categorical policy of discrete action space, as stated in eq 16, $\frac{\partial V_{jt}^{R}}{\partial H(\pi^i)} = \frac{\partial V_{jt}^{R}}{\partial V_i^R} \times \frac{\partial V_i^R}{\partial H(\pi^i)}$. The second term is approximated by $\frac{\Delta V_i^R(\tau^i)}{\Delta \mathcal{H}(\pi^i_t)}$. However, it is still unclear to me how it is computed numerically.

**Summary Of The Paper:**

This paper presents an MARL algorithm to adpatively handle the entropy regularization in multi-agent RL. In this method, the level of exploration of each agent is controlled by its time-varying target entropy, which severs as a constraint in the optimization problem. To determine a proper target entropy for each agent, the authors propose a partial derivative-based metric to evaluate the benefit of exploration and then use this term to decide the target entropy for each agent. Empirically, in a customized SMAC environment, their method is demonstrated to be better than other value-decomposition methods.


**Summary Of The Review:**

This paper studies adaptive entropy regularization in MARL, and provides a naive yet reasonable solution for this problem. However, the presentation of this paper can be substantially improved, and some technical details about the computation of partial derivatives have to be further explained.  In short, I will not recommend acceptance of this paper, but I will be happy to reconsider it if the authors can address all those problems.

---

> ### Author Response · Authors · 2022-11-16
> **Response to Reviewer mfnN (1)**
>
> <Regarding the writing and presentation>
>
> - For the given short period of revision time, we revised the paper.  Now, we summarized the algorithm as Algorithm 1 in Appendix B.2 and presented the overall learning architecture separately for continuous and discrete action cases in Figures 6 and 7 in Appendix B.2.  We also included a detailed explanation on the computation of the proposed metric, which is an important issue in the proposed method, in Appendix B.1.
>
> - We will revise the paper further to improve readability and clarity.
>
>
> <Regarding the standard SMAC tasks and more baselines>
>
> - We have added additional environments on the standard SMAC environment.
> - We have included MAPPO as a baseline on the sparse SMAC environment.
> - We have included FOP as a baseline for the continuous cooperative matrix game. It is shown that FOP fails to learn to reach the target circle, which is the goal of the environment. This is because 1) FOP learns the individual temperature parameter to maximize the cumulative expected return by enhancing value factorization, not for \textit{multi-agent exploration-exploitation trade-off}, and 2) FOP uses the same value for all individual temperature parameters for learning policies. Thus, indeed ADER is the first algorithm that considers the multi-agent exploration-exploitation tradeoff.
>
> <Q1: Regarding the computation of the state value functions>
>
> - Sorry for the confusion. The definition of the state value function regarding the return is  $V^R_{JT}(s_t,\mathbf{\tau_t}) = \mathbb{E}{a_t}  [Q^R_{JT}(s_t,\mathbf{\tau_t}, \mathbf{a_t})]$ , i.e., the marginalization of $Q^R_{JT}$  over the joint action and this is used for theoretical proof of Proposition 1.  However, the practical computation of $V_{JT}^R$ based on  the marginalization of $Q^R_{JT}$  over the joint action is difficult because the combination of joint actions increases exponentially as a function of the number of agents.  So, for the practical computation of $V_{JT}^R$, we marginalize the individual $Q$-function based on individual action to get $V_i^R$. Then, we feed $V_1^R, \cdots, V_N^R$ of all agents to the mixing network $f_{mix}^{V,R}$ to obtain the joint state value function as
> $V_{JT}^R(s,\mathbf{\tau}) = f_{mix}^{V,R}(s, V^R_1(\tau^1), \cdots, V^R_N(\tau^N))$. Here,  $f_{mix}^{V,R}$ is learned such that  $f_{mix}^{V,R}$ follows the definition by the TD loss eq. (B.2) and the Bellman equation.
>
>
>
>
> <Q2: Regarding the individual state-action value function>
>
> - The sum of the mixing network and individual value networks can be viewed as a big single neural network.  Value decomposition imposes a structure on this single big network as the sum of  the mixing network and individual value networks.  It is known that monotonic value factorization of the overall value network yields better learning of the value function and final performance in MARL by properly restricting the representation of the joint value function  [Ref. 3-1].
>
>
> [Ref. 3-1] Rashid, Tabish, et al. "Qmix: Monotonic value function factorisation for deep multi-agent reinforcement learning." International conference on machine learning. PMLR, 2018.

---

> ### Author Response · Authors · 2022-11-16
> **Response to Reviewer mfnN (2)**
>
> <Q3: Regarding the computation of the proposed metric>
>
> - Sorry for the previous too brief explanation on the computation of the metric. We agree on that the computation of the metric in the first submission is too brief. Now we have added a detailed explanation on the computation of the metric in Appendix B.1. The explanation is as follows.
>
>
> We adopted an actor-critic structure for our algorithm. Hence,  for each agent we have a separate actor, i.e., policy in both continuous-action and discrete-action cases, as seen in Figures 6 and 7 in Appendix B,  which show the overall structure for continuous-action and discrete-action cases, respectively.
> The computation of   the partial derivative   $\frac{\partial V_{JT}^R(s,\mathbf{\tau})}{\partial \mathcal{H}(\pi_t^i(\cdot|\tau^i))}$
> depends on the overall structure, especially on the structure of the individual critic network.
>
> First, consider the continuous-action case. In this case, we used a Gaussian policy for each agent. Then, the policy neural network of Agent $i$ with trainable parameter $\theta^i$
> takes trajectory $\tau_t^i$ as input and generates the mean $\mu^i$ and the log variance $\log \sigma^i$ as output, as shown in Figure 6 in Appendix B.   Based on these outputs and the reparameterization trick, the action of Agent $i$ is generated as $a^i = \mu^i + \exp (\log \sigma^i) Z^i$, where $Z^i$ is Gaussian-distributed with zero mean and identity covariance matrix, i.e., $Z^i \sim N(0, I)$.
> The action $a^i$ and observation $\tau^i$ are applied as input to both return and entropy critic networks for Agent $i$, as seen in Figure 6 in Appendix B. Now, focus on the return critic network of Agent $i$, which is relevant to the computation of our metric. The return critic of Agent $i$ generates the local Q-value $Q_i^R(\tau^i,a^i)$.  All local $Q$-values $Q_1^R(\tau^1,a^1), \cdots, Q_N^R(\tau^N,a^N)$ from all agents are applied as input to the mixing network for global return value $Q_{JT}^R$, as seen in Figure 6 in Appendix B. Due to the connected tensor structure in  Figure 6 in Appendix B, at the time of learning, the gradient of $Q_{JT}^R$ with respect to $\log \sigma^i$ can be computed by deep learning libraries such as Pytorch. Note that $\log \sigma^i$ is simply a scaled version of the Gaussian policy entropy. So, we can just obtain this value $\partial Q_{JT}^R/ \partial \log \sigma^i$ from deep learning libraries. Furthermore, $V_{JT}^R$ can be obtained by sampling multiple $a^i$'s from the same policy $\pi_t^i$, computing the corresponding multiple $Q$-values and taking the average over the multiple $a^i$ samples. However, we simplify this step and just use $\partial Q_{JT}^R/ \partial \log \sigma^i$ as our estimate for the metric $\frac{\partial V_{JT}^R(s,\mathbf{\tau})}{\partial \mathcal{H}(\pi_t^i(\cdot|\tau^i))}$.  Indeed, many algorithms use single-sample average for obtaining expectations for algorithm simplicity.
>
>
> Second, consider the discrete-action case. In this case, we again use an actor-critic structure for our algorithm.
> The structure of the critic network of Agent $i$ in the discrete-action case is different from that in the continuous-action case. Whereas the critic network takes the observation $\tau^i$ and the action $a^i$ as input,  and generates $Q_i^R(\tau^i,a^i)$ in the continuous-action case,   the critic network typically uses the DQN structure (Mnih et al. 2015), which takes  the observation $\tau^i$ as input and generates  all  $Q_i^R(\tau^i,a^i_1), \cdots, Q_i^R(\tau^i,a^i_{|\mathcal{A}|})$ as output in the discrete-action case.
> In the discrete-action case, action is over a finite action set $\mathcal{A}=\{a_1,\cdots,a_{|\mathcal{A}|}\}$, and the policy is described by a categorical distribution $\mathbf{p}^i=\left[p^i_1,\cdots,p^i_{|\mathcal{A}|}\right]$ over $\mathcal{A}$  for each  state (or trajectory).  Hence, our actor, i.e, policy $\pi^i$ for Agent $i$ is a deep neural network which takes  the observation $\tau^i$ as input and generates  probability vector $\mathbf{p}^i=\left[p^i_1,\cdots,p^i_{|\mathcal{A}|}\right]$ as output. Here, let us denote the policy deep neural network parameter by $\theta^i$ and denote the policy $\pi_t^i$ by $\pi_{\theta^i}^i$, showing the current parameter explicitly.  Then, using the output $\mathbf{p}^i=\left[p^i_1,\cdots,p^i_{|\mathcal{A}|}\right]$
>  of the policy network and the output $Q_i^R(\tau^i,a^i_1), \cdots, Q_i^R(\tau^i,a^i_{|\mathcal{A}|})$ of the critic network, we compute the local return value  as
>
>
> $V_i^R(\tau^i)=\sum_{j=1}^{|\mathcal{A}|} p^i_j(\tau^i) Q_i^R(\tau^i, a^i_j). ~~~~~~~  (1) $
>
> Then, all local return values $V_1^R(\tau^1),\cdots,V_N^R(\tau^N)$ are fed to the mixing network for global return value $V_{JT}^R$, as seen in Figure 7 of Appendix B.

---

> ### Author Response · Authors · 2022-11-16
> **Response to Reviewer mfnN (3)**
>
> <Q3: Regarding the computation of the proposed metric (cont) >
>
>
>
> In this discrete-action case,
> the policy entropy is given by
>  $\mathcal{H}(\pi_{\theta^i}^i(\cdot|\tau^i))=-\sum_{j=1}^{N}p^i_j \log p^i_j$.  On the contrary to the continuous-action case in which the policy entropy $\log \sigma^i$ is an explicit node value in the overall structure and hence the output $V_{JT}^R$ gradient with respect to the node $\log \sigma^i$ is directly available,  in the discrete-action case there is no node corresponding to the value  $\mathcal{H}(\pi_{\theta^i}^i(\cdot|\tau^i))=-\sum_{j=1}^{N}p^i_j \log p^i_j$. Hence, the gradient $\frac{\partial V_{JT}^R}{\partial \mathcal{H}(\pi_{\theta^i}^i)}$ is not readily available from the architecture.  Note that we only have nodes for $p^i_1,\cdots,p^i_{|\mathcal{A}|}$ in the architecture, but the gradient of $V_{JT}^R$ with respect to $p^i_j$ is not $\frac{\partial V_{JT}^R}{\partial \mathcal{H}(\pi_{\theta^i}^i)}$. Furthermore, it is not easy to compute $\frac{\partial V_{JT}^R}{\partial \mathcal{H}(\pi_{\theta^i}^i)}$ from $\frac{\partial V_{JT}^R}{ \partial p^i_j}$,  $j=1,\cdots, |\mathcal{A}|$ with $\sum_j p^i_j=1$ for general cardinality $|\mathcal{A}|$.
>
>
> To circumvent this difficulty and compute the metric $\frac{\partial V_{JT}^R}{\partial \mathcal{H}(\pi_{\theta^i}^i)}$, we exploit  the policy network parameter $\theta^i$ and numerical computation. When the current policy network parameter is $\theta^i$, we have the corresponding policy network output $p^i_1,\cdots,p^i_{|\mathcal{A}|}$. Then, consider the temporary scalar objective function $\mathcal{H}(\pi_{\theta^i}^i)$ for the policy network. We can compute the gradient of $\mathcal{H}(\pi_{\theta^i}^i)$ with respect to the policy parameter $\theta^i$. Let us denote this gradient by $\frac{\partial \mathcal{H}(\pi_{\theta^i}^i)}{\partial \theta^i}$, which is the direction of $\theta^i$ for maximum policy entropy increase. Then, we update the policy parameter as
> $\tilde{\theta}^i = \theta^i + \delta \frac{\partial \mathcal{H}(\pi_{\theta^i}^i)}{\partial \theta^i}$,
> where $\delta$ is a positive stepsize. Then, for the updated policy $\pi^i_{\tilde{\theta}^i}$, we compute the corresponding $p^i_1,\cdots,p^i_{|\mathcal{A}|}$. Using these updated probability values, we compute the local value $V_i^R$ by using eq. (1).  Using the values before and after the update, we compute
> $\frac{\Delta V^R_i(\tau^i)}{\Delta \mathcal{H}(\pi^i)} = \frac{V^R_i(\tau^i;{\pi}^i_{\tilde{\theta}^i})-V^R_i(\tau^i;\pi^i_{\theta^i})}{\mathcal{H}({\pi}^i_{\tilde{\theta}^i})-\mathcal{H}(\pi^i_{\theta^i})}$.
>
> Now, the metric $\frac{\partial V_{JT}^R}{\partial \mathcal{H}(\pi_{\theta^i}^i)}$ can be computed based on the chain rule. That is, we have
> $\frac{\partial V_{JT}^R}{\partial \mathcal{H}(\pi_{\theta^i}^i)}=\frac{\partial V_{JT}^R(s,\mathbf{\tau})}{\partial V_i^R(\tau^i)}\times \frac{\partial V_i^R(\tau^i)}{\partial \mathcal{H}(\pi_{\theta^i}^i)}$. Here, the first term  $\frac{\partial V_{JT}^R(s,\mathbf{\tau})}{\partial V_i^R(\tau^i)}$ is available from deep learning libraries since $V_{JT}^R$ and $V_i^R$ are nodes of the learning architecture. The second term $\frac{\partial V_i^R(\tau^i)}{\partial \mathcal{H}(\pi_{\theta^i}^i)}$ can be approximated by  $\frac{\Delta V_i^R(\tau^i)}{\Delta \mathcal{H}(\pi^i)}$ in the above.
>
> Note that the policy update $\tilde{\theta}^i = \theta^i + \delta \frac{\partial \mathcal{H}(\pi_{\theta^i}^i)}{\partial \theta^i}$ is only for computation of the metric. It is not done for the actual learning update.
>
> In fact, this technique of computing $\frac{\partial V_{JT}^R}{\partial \mathcal{H}(\pi_{\theta^i}^i)}$ can be useful from other aspects in entropy-based RL and is our contribution.
>
> This detailed explanation has been included in Appendix B.1 of the revised paper.

---

### Official Review · Reviewer_dyHK · 2022-10-23

**Confidence:** 4
**Correctness:** 4
**Technical Novelty And Significance:** 4
**Empirical Novelty And Significance:** 3
**Recommendation:** 8

**Clarity, Quality, Novelty And Reproducibility:**

The paper has good clarity overall. Most of the key components of ADER are well motivated and clearly explained, although the necessity of using monotonic value function factorization is not very clear to me.

The key idea of ADER looks novel to me.

**Strength And Weaknesses:**

- Strengths:
	- The proposed framework is well motivated and seems to work well in a variety of complex cooperative tasks that require adaptive exploration across agents over time.
	- The main idea of adaptively learning an individual target entropy for each agent over time to better balance the exploration and exploitation in the dimension of agents is very interesting and seems novel to me.
	- The paper is well-written and easy to follow overall.
- Weaknesses:
	- It is not clear how the proposed method performs in cooperative tasks that do not require adaptive exploration across agents over time.

**Summary Of The Paper:**

This paper proposes an adaptive entropy-regularization framework (ADER) to address the multi-agent exploration-exploitation trade-off problem in MARL. The key insight of ADER is that the amount of exploration every agent needs to perform is different and can change over time, so we need to adaptively control the amount of exploration each agent conducts and learn this amount across agents and over time. To achieve this, ADER adaptively learn an individual target entropy for each agent over time (to control the amount of exploration for each agent), assuming a fixed total entropy budget. And it uses the change in the joint pure-return value w.r.t. the change in agent $i$'s policy entropy to estimate the benefits of increasing the target entropy of agent $i$.  In addition, in ADER, the exploration and exploitation is disentangled by disentangling the return from the entropy (i.e., the joint soft Q-function is decomposed into one joint Q-function for reward and one joint Q-function for the entropy).

**Summary Of The Review:**

I like the main idea of ADER. It seems to provide a simple yet effective way to better coordinate the level of exploration across different agents over time in cooperative tasks. I think the authors do a very good job at explaining the main intuitions of the proposed method. For instance, the motivation example is quite nice. It clearly explains how one agent's exploration can change over time and how it can hinder other agents' exploitation, thus showing the necessity of adaptively controlling the trade-off between exploitation and exploration across multiple agents.

My only (small) concern of the paper is most of the cooperative tasks tested (e.g., predator-prey and SMAC) in the experiments were different from the original settings. They were specifically designed to be difficult for the agents to gain positive rewards under simultaneous exploration (or similar levels of exploration). While I understand this might be because exploration is not a big problem in these original tasks, it seems that the authors could have evaluated their method on MARL benchmark like Google Research Football (with the original setting), which is known to have hard exploration issues. Also, it'd be nice to know if ADER could perform well in the original SMAC tasks, such that we could know if continuously learning the target entropy for each agent could hurt performance in cooperative tasks that do not require adaptive exploration across agents.

Some minor comments/suggestions:
- I think the introduction could be improved some. For example, the introduction says "This disentanglement alleviates instability which can occur due to the updates of the temperature parameters and enables applying value factorization to return and entropy separately." It suddenly mentions value function factorization, but does not explain the motivation/necessity for using it.
- The related work section mentions that LICA does not maximize the cumulative sum of entropy but regularize the action entropy. Why maximizing the cumulative sum of entropy can be better than regularizing the action entropy? This could be explained some.
- The ablation study section mentions "we compare ADER with and without the monotonic constraint to show the necessity of the monotonic constraint." For ADER without the monotonic constraint, how does it learn the joint $Q$ or $V$? Does it learn a centralized critic that directly conditions on the global state and joint action of agents? Also, I do not understand why it is necessary to use a *monotonic* value factorization. ADER learns both actors and critics and only the actors are needed during execution. So one can also use a *nonmonotonic* value function factorization.

---

> ### Author Response · Authors · 2022-11-16
> **Response to Reviewer dyHK**
>
> <Regarding additional experiments on Google Research Football and the original SMAC tasks>
>
> - As the reviewer recommended, we have added experiments on the original SMAC environment and google research football environment in Appendix D. It is seen that ADER significantly outperforms the previous baselines.
>
>
> <Q1: Regarding the writing and presentation>
>
> - For the given short period of revision time, we revised the paper.  Now, we summarized the algorithm as Algorithm 1 in Appendix B.2 and presented the overall learning architecture separately for continuous and discrete action cases in Figures 6 and 7 in Appendix B.2.  We also included a detailed explanation on the computation of the proposed metric, which is an important issue in the proposed method, in Appendix B.1.
>
> - We will revise the paper further to improve readability and clarity.
>
>
>
>
> <Q2:  Regarding maximizing the cumulative sum entropy>
>
> - Regularizing the current action entropy encourages the agent to do more random actions at each time step.  However, the maximum entropy RL does not do this. Instead, maximum entropy RL such as the SAC algorithm includes the policy entropy as a reward and maximized the cumulative sum entropy of the entire trajectory.  This encourages the agent to explore a path on which it visits states with high action diversity. It is known that this yields better exploration.
>
> <Q3: Regarding the monotonic value factorization>
>
> -  Value factorization without monotonic constraint:
>  The monotone constraint was implemented with a positive-weight mixing network.
> To make the weights in the mixing network positive, absolute activation functions such as ReLU are considered in QMIX.  We adopted this method to implement a positive-weight mixing network. For a mixing network without the monotone constraint, we can simply
> use the tangent hyperbolic function instead of absolute activation functions. This is the only change. The learning is the same as the case with the monotone constraint except for the change of the activation function in the mixing network.
>
> - Benefit of the monotone constraint:  As aforementioned,
>  we can use value function factorization without monotonic constraint. However, generally value function factorization without monotone constraint is used. This is because it is known that value factorization with monotone constraint yields better performance than value factorization without monotonic constraint [Ref. 2-1, Ref. 2-2]
>
>
> [Ref. 2-1] Rashid, Tabish, et al. "Qmix: Monotonic value function factorisation for deep multi-agent reinforcement learning." International conference on machine learning. PMLR, 2018.
>
> [Ref. 2-2] Su, Jianyu, Stephen Adams, and Peter Beling. "Value-decomposition multi-agent actor-critics." Proceedings of the AAAI Conference on Artificial Intelligence. Vol. 35. No. 13. 2021.

---

### Official Review · Reviewer_5qy9 · 2022-10-24

**Confidence:** 3
**Correctness:** 4
**Technical Novelty And Significance:** 3
**Empirical Novelty And Significance:** 2
**Recommendation:** 6

**Clarity, Quality, Novelty And Reproducibility:**

This paper is well-written with high clarity. Somewhat novel, but not groundbreakingly novel. I think the authors showed good amount of experiments and evaluations on various benchmarks, and ablation studies, which seem to be reproducible.

**Strength And Weaknesses:**

- The strength of the paper comes from the idea that, while previous works encourage same level of exploration across agents, this work proposes to differentiate the level of exploration across agents in multi-agent RL setting.
- Another strength comes from the core idea of this work: joint soft value function decomposition / separated factorization.
- The motivation part 3.1 sounds convincing to me; one agent’s exploration can hinder other agent’s exploitation, resulting that simultaneous exploration of multiple agents can make learning unstable. Need a framework that can adaptively learn proper levels of exploration for each agent.
- Experiments are well done, not extensive though.

Questions.

Q1. Question about ADER performance shown in Figure 2a. It seems that ADER outperforms other methods like SER-DCE, SER-MARL, but there is a point where ADER’s performance suddenly jumps up in the middle. Is there any explanation on why this happens?

Q2. In Appendix B, could you give me more justification on setting the coefficient beta_i? Especially, line B.7, beta_i are defined as softmax of expectation of \partial V^R_{JT}(s,\tau) / \partial H (pi (|))) ? Could you give us more detailed explanations on it? And can you explain why it is difficult to directly obtain the partial derivative in discrete-action case, and using chain rule is justified?

**Summary Of The Paper:**

This paper aims to solve the exploration-exploitation tradeoff problem in the context of multi-agent reinforcement learning. While there have been a myriad of works on exploration-exploitation tradeoff on single age reinforcement learning, there are not many on the multi agent RL. This work proposes an adaptive entropy-regularization framework that learns adequate amount of exploration for each agent. To this end, this work proposes to decompose the joint soft value function into pure return and entropy sum. This disentanglements enable a more stable while updating the temperature parameters. This work focuses entropy-based MARL.

**Summary Of The Review:**

I would give marginally above the acceptance threshold. It would be good if the authors could answer my questions. There might be some issues that I didn’t catch, and if other reviewers have raised issues, I’m happy to discuss.

---

> ### Author Response · Authors · 2022-11-16
> **Response to Reviewer 5qy9 (1)**
>
> <Regarding the jump in the performance of ADER in Figure 2>
>
>
> The jump is because of our reward design.  We modified the reward function of the continuous matrix game considered in [Ref. 1-1].  The reward function in the first subpath is
> $r = x + max(0.1, 0.00001 / (abs(x - 0.6) + 0.00001))$. This reward function increases linearly before the neighborhood of the end of the first subpath ($x=0.6,y=0$). In the neighborhood of  the end of the first subpath ($x=0.6,y=0$), the reward function suddenly jumps to $1$. The reward in the second path increases up to approximately 3 at the end of the second subpath. Finally, the maximum reward $5$ is obtained when the position reaches the center of the target circle.
>
> One episode for this matrix game consists of just one time step. The action is $(a_1,a_2)$. The agent generates an action according to the policy  $a_1 \sim N(\mu_1, \sigma_1^2)$ and  $a_2 \sim N(\mu_2, \sigma_2^2)$, where the policy parameters $\mu_1,\mu_2,\sigma_1^2$ and $\sigma_2^2$ are updated after each time step based on the reward. The reward is obtained when the endpoint of the vector $(a_1,a_2)$ is inside the rotated L shape path.
>
> For simple explanation, let us simplify the action as taking a point $(a_1,a_2)$ uniformly within an ellipse with center $(\mu_1,\mu_2)$ and eccentricity $\frac{\sigma_2}{\sigma_1}$, although Gaussianness can be added for exact description. In this simplified version,  the policy can be described by this ellipse.
>
> The initial policy is $\mu_1=\mu_2=0$ and $\sigma_1=\sigma_2$ and hence the initial policy is a small circle centered at (0,0). After taking a few actions, the agent knows that from the left-side pointing vectors the agent gets negative rewards and from the right-side pointing vectors the agent gets positive rewards when the point vector is aligned with the positive $x$ direction. This is because the positive reward is available only on the rotated L-shaped path. Thus, the agents quickly learn to make the policy an x-aligned ellipse and move the policy ellipse to the right side. This corresponds to 0 to 3e4 time steps. Please see Fig. 2(b). In  Fig. 2(b), the target entropy values are $\log \sigma_1$ and $\log \sigma_2$. Hence, we can estimate the
>  eccentricity $\frac{\sigma_2}{\sigma_1}$ of the ellipse from the two values $\log \sigma_1$ and $\log \sigma_2$.
> We also observed this in the $(\mu_1,\mu_2)$ plot over time step.  Once the whole $x$-range of the  policy ellipse is  within $x\ge 0$,  left-pointing action vectors and right-pointing action vectors do not make a big difference in reward for a small policy ellipse. Hence, the agent tries even outside of the $x$-axis for searching reward, so the ellipse becomes a bit fat, i.e.,  eccentricity $\frac{\sigma_2}{\sigma_1}$ is a bit increased. This corresponds to the time steps 3e4 to 7e4 time steps. Please see Fig. 2(b). But, due to the small slope of the reward function along the $x$-axis, the policy ellipse learns to slowly move to the right side. This corresponds to the time step 7e4 to 16e4 in Fig. 2(b). Once the right side of the policy ellipse hits the high reward neighborhood of $(0.6,0)$, the agent knows that it receives a high reward from the right side. Hence, it quickly moves to the right side and makes the ellipse flatter again, i.e.  decreasing the eccentricity $\frac{\sigma_2}{\sigma_1}$.   This corresponds to the time step 16e4 to 19e4 in Fig. 2(b). Once the policy ellipse is around (0.6,0), it learns that it can get even higher reward from the second path from (0.6,0) to (0.6,0.6). The policy ellipse then climbs up the second path. Since the reward gradient along the second path is larger, it quickly climbs up to the target point.  So, from time step 19e4, the reward quickly reaches the maximum 5.
>
> Indeed, ADER shows the expected behavior following our reward function design.
>
> [Ref. 1-1] Peng, Bei, et al. "Facmac: Factored multi-agent centralised policy gradients." Advances in Neural Information Processing Systems 34 (2021): 12208-12221.

---

> ### Author Response · Authors · 2022-11-16
> **Response to Reviewer 5qy9 (2)**
>
> <Regarding the justification on setting the coefficient $\beta_i$>
>
> We aim to allocate higher (or lower) target entropy to agents whose $\partial V_{JT}^R/\partial \mathcal{H}(\pi^i_t)$ is larger (or smaller) than that of other agents. For this, we first compute the values of $\partial V_{JT}^R/\partial \mathcal{H}(\pi^i_t)$ of all agents and assign the target entropy for each agent as a monotone function of this metric. Note that the total target entropy budget is $\mathcal{H}_0$ and so this should be distributed as $\mathcal{H}_1, \cdots, \mathcal{H}_N$ to Agents $1, \cdots, N$, respectively such that  $\mathcal{H}_1 + \cdots + \mathcal{H}_N=\mathcal{H}_0$.
>
> This can be done by introducing $\beta_1,\cdots,\beta_N$ such that $\sum_{i=1}^N
> \beta_i =1$ and $\beta_i \ge 0,\forall i$, and setting the target entropy of Agent $i$ as $\mathcal{H}_i = \beta_i \times \mathcal{H}_0$ when $\mathcal{H}_0\ge 0$.
>
> Here, $\beta_i$ should be an increasing function of $\mathbb{E}\Big[\frac{\partial V_{JT}^R(s,\mathbf{\tau})}{\partial \mathcal{H}(\pi_t^i(\cdot|\tau^i))}\Big]$.
> One way to satisfy these conditions is to use the softmax function (note that the outputs of a softmax function are nonnegative and sum up to one.) Hence, if we set
> \begin{equation}
>   \small{ \big[\beta_1,  \cdots, \beta_i, \cdots, \beta_N \big]=     \mbox{Softmax}\Bigg[\mathbb{E}\Big[\frac{\partial V_{JT}^R(s,\mathbf{\tau})}{\partial \mathcal{H}(\pi_t^1(\cdot|\tau^1))} \Big],\cdots, \mathbb{E}\Big[\frac{\partial V_{JT}^R(s,\mathbf{\tau})}{\partial \mathcal{H}(\pi_t^i(\cdot|\tau^i))}\Big], \cdots, \mathbb{E}\Big[\frac{\partial V_{JT}^R(s,\mathbf{\tau})}{\partial \mathcal{H}(\pi_t^N(\cdot|\tau^N))}\Big] \Bigg]}
> \end{equation}
> and $\mathcal{H}_i = \beta_i \times \mathcal{H}_0$, then the required properties are all satisfied.

---

> ### Author Response · Authors · 2022-11-16
> **Response to Reviewer 5qy9 (3)**
>
> <Regarding the computation of the proposed metric in discrete-action cases>
>
>
>
>
> Sorry for the previous too brief explanation on the computation of the metric. We agree that the computation of the metric in the first submission is too brief. Now we have added a detailed explanation on the computation of the metric in Appendix B.1. The explanation is as follows.
>
>
> We adopted an actor-critic structure for our algorithm. Hence,  for each agent we have a separate actor, i.e., policy in both   continuous-action and discrete-action cases, as seen in Figures 6 and 7 in Appendix B,  which show the overall structure for continuous-action and discrete-action cases, respectively.
> The computation of   the partial derivative   $\frac{\partial V_{JT}^R(s,\mathbf{\tau})}{\partial \mathcal{H}(\pi_t^i(\cdot|\tau^i))}$
> depends on the overall structure, especially on the structure of the individual critic network.
>
>
> First, consider the continuous-action case. In this case, we used a Gaussian policy for each agent. Then, the policy neural network of Agent $i$ with trainable parameter $\theta^i$
> takes trajectory $\tau_t^i$ as input and generates the mean $\mu^i$ and the log variance $\log \sigma^i$ as output, as shown in Figure 6 in Appendix B.   Based on these outputs and the reparameterization trick, the action of Agent $i$ is generated as $a^i = \mu^i + \exp (\log \sigma^i) Z^i$, where $Z^i$ is Gaussian-distributed with zero mean and identity covariance matrix, i.e., $Z^i \sim N(0, I)$.
> The action $a^i$ and observation $\tau^i$ are applied as input to both return and entropy critic networks for Agent $i$, as seen in Figure 6 in Appendix B. Now, focus on the return critic network of Agent $i$, which is relevant to the computation of our metric. The return critic of Agent $i$ generates the local Q-value $Q_i^R(\tau^i,a^i)$.  All local $Q$-values $Q_1^R(\tau^1,a^1), \cdots, Q_N^R(\tau^N,a^N)$ from all agents are applied as input to the mixing network for global return value $Q_{JT}^R$, as seen in Figure 6 in Appendix B. Due to the connected tensor structure in  Figure 6 in Appendix B, at the time of learning, the gradient of $Q_{JT}^R$ with respect to $\log \sigma^i$ can be computed by deep learning libraries such as Pytorch. Note that $\log \sigma^i$ is simply a scaled version of the Gaussian policy entropy. So, we can just obtain this value $\partial Q_{JT}^R/ \partial \log \sigma^i$ from deep learning libraries. Furthermore, $V_{JT}^R$ can be obtained by sampling multiple $a^i$'s from the same policy $\pi_t^i$, computing the corresponding multiple $Q$-values and taking the average over the multiple $a^i$ samples. However, we simplify this step and just use $\partial Q_{JT}^R/ \partial \log \sigma^i$ as our estimate for the metric $\frac{\partial V_{JT}^R(s,\mathbf{\tau})}{\partial \mathcal{H}(\pi_t^i(\cdot|\tau^i))}$.  Indeed, many algorithms use single-sample average for obtaining expectations for algorithm simplicity.
>
> Second, consider the discrete-action case. In this case, we again use an actor-critic structure for our algorithm.
> The structure of the critic network of Agent $i$ in the discrete-action case is different from that in the continuous-action case. Whereas the critic network takes the observation $\tau^i$ and the action $a^i$ as input,  and generates $Q_i^R(\tau^i,a^i)$ in the continuous-action case,   the critic network typically uses the DQN structure (Mnih et al. 2015), which takes  the observation $\tau^i$ as input and generates  all  $Q_i^R(\tau^i,a^i_1), \cdots, Q_i^R(\tau^i,a^i_{|\mathcal{A}|})$ as output in the discrete-action case.
> In the discrete-action case, action is over a finite action set $\mathcal{A}=\{a_1,\cdots,a_{|\mathcal{A}|}\}$, and the policy is described by a categorical distribution $\mathbf{p}^i=\left[p^i_1,\cdots,p^i_{|\mathcal{A}|}\right]$ over $\mathcal{A}$  for each  state (or trajectory).  Hence, our actor, i.e, policy $\pi^i$ for Agent $i$ is a deep neural network which takes  the observation $\tau^i$ as input and generates  probability vector $\mathbf{p}^i=\left[p^i_1,\cdots,p^i_{|\mathcal{A}|}\right]$ as output. Here, let us denote the policy deep neural network parameter by $\theta^i$ and denote the policy $\pi_t^i$ by $\pi_{\theta^i}^i$, showing the current parameter explicitly.  Then, using the output $\mathbf{p}^i=\left[p^i_1,\cdots,p^i_{|\mathcal{A}|}\right]$
>  of the policy network and the output $Q_i^R(\tau^i,a^i_1), \cdots, Q_i^R(\tau^i,a^i_{|\mathcal{A}|})$ of the critic network, we compute the local return value  as
>
>
> $V_i^R(\tau^i)=\sum_{j=1}^{|\mathcal{A}|} p^i_j(\tau^i) Q_i^R(\tau^i, a^i_j).  ~~~~~~~~~~~~ (1)$
>
>
>
> Then, all local return values $V_1^R(\tau^1),\cdots,V_N^R(\tau^N)$ are fed to the mixing network for global return value $V_{JT}^R$, as seen in Figure 7 of Appendix B.

---

> ### Author Response · Authors · 2022-11-16
> **Response to Reviewer 5qy9 (4)**
>
> <Regarding the computation of the proposed metric in discrete-action cases (cont)>
>
> In this discrete-action case,
> the policy entropy is given by
>  $\mathcal{H}(\pi_{\theta^i}^i(\cdot|\tau^i))=-\sum_{j=1}^{N}p^i_j \log p^i_j$.  On the contrary to the continuous-action case in which the policy entropy $\log \sigma^i$ is an explicit node value in the overall structure and hence the output $V_{JT}^R$ gradient with respect to the node $\log \sigma^i$ is directly available,  in the discrete-action case there is no node corresponding to the value  $\mathcal{H}(\pi_{\theta^i}^i(\cdot|\tau^i))=-\sum_{j=1}^{N}p^i_j \log p^i_j$. Hence, the gradient $\frac{\partial V_{JT}^R}{\partial \mathcal{H}(\pi_{\theta^i}^i)}$ is not readily available from the architecture.  Note that we only have nodes for $p^i_1,\cdots,p^i_{|\mathcal{A}|}$ in the architecture, but the gradient of $V_{JT}^R$ with respect to $p^i_j$ is not $\frac{\partial V_{JT}^R}{\partial \mathcal{H}(\pi_{\theta^i}^i)}$. Furthermore, it is not easy to compute $\frac{\partial V_{JT}^R}{\partial \mathcal{H}(\pi_{\theta^i}^i)}$ from $\frac{\partial V_{JT}^R}{ \partial p^i_j}$,  $j=1,\cdots, |\mathcal{A}|$ with $\sum_j p^i_j=1$ for general cardinality $|\mathcal{A}|$.
>
>
> To circumvent this difficulty and compute the metric $\frac{\partial V_{JT}^R}{\partial \mathcal{H}(\pi_{\theta^i}^i)}$, we exploit  the policy network parameter $\theta^i$ and numerical computation. When the current policy network parameter is $\theta^i$, we have the corresponding policy network output $p^i_1,\cdots,p^i_{|\mathcal{A}|}$. Then, consider the temporary scalar objective function $\mathcal{H}(\pi_{\theta^i}^i)$ for the policy network. We can compute the gradient of $\mathcal{H}(\pi_{\theta^i}^i)$ with respect to the policy parameter $\theta^i$. Let us denote this gradient by $\frac{\partial \mathcal{H}(\pi_{\theta^i}^i)}{\partial \theta^i}$, which is the direction of $\theta^i$ for maximum policy entropy increase. Then, we update the policy parameter as
> $\tilde{\theta}^i = \theta^i + \delta \frac{\partial \mathcal{H}(\pi_{\theta^i}^i)}{\partial \theta^i}$,
> where $\delta$ is a positive stepsize. Then, for the updated policy $\pi^i_{\tilde{\theta}^i}$, we compute the corresponding $p^i_1,\cdots,p^i_{|\mathcal{A}|}$. Using these updated probability values, we compute the local value $V_i^R$ by using eq. (1).  Using the values before and after the update, we compute $\frac{\Delta V_i^R(\tau^i)}{\Delta \mathcal{H}(\pi^i)}$ =  $\frac{V_i^R(\tau^i; {\pi}^i_{\tilde{\theta}^i}) -V_i^R(\tau^i;\pi^i_{\theta^i})}{\mathcal{H}({\pi}^i_{\tilde{\theta}^i})-\mathcal{H}(\pi^i_{\theta^i})}. $
>
>
> Now, the metric $\frac{\partial V_{JT}^R}{\partial \mathcal{H}(\pi_{\theta^i}^i)}$ can be computed based on the chain rule. That is, we have
> $\frac{\partial V_{JT}^R}{\partial \mathcal{H}(\pi_{\theta^i}^i)}=\frac{\partial V_{JT}^R(s,\mathbf{\tau})}{\partial V_i^R(\tau^i)}\times \frac{\partial V_i^R(\tau^i)}{\partial \mathcal{H}(\pi_{\theta^i}^i)}$. Here, the first term  $\frac{\partial V_{JT}^R(s,\mathbf{\tau})}{\partial V_i^R(\tau^i)}$ is available from deep learning libraries since $V_{JT}^R$ and $V_i^R$ are nodes of the learning architecture. The second term $\frac{\partial V_i^R(\tau^i)}{\partial \mathcal{H}(\pi_{\theta^i}^i)}$ can be approximated by  $\frac{\Delta V_i^R(\tau^i)}{\Delta \mathcal{H}(\pi^i)}$ in the above.
>
> Note that the policy update $\tilde{\theta}^i = \theta^i + \delta \frac{\partial \mathcal{H}(\pi_{\theta^i}^i)}{\partial \theta^i}$ is only for computation of the metric. It is not done for the actual learning update.
>
> In fact, this technique of computing $\frac{\partial V_{JT}^R}{\partial \mathcal{H}(\pi_{\theta^i}^i)}$ can be useful from other aspects in entropy-based RL and is our contribution.
>
>
> This detailed explanation has been included in Appendix B.1 of the revised paper.

---

### Author Response · Authors · 2022-11-16
**Common response**

We thank all reviewers for their valuable comments.
In this paper, we propose an adaptive entropy-regularization framework called ADER to address the multi-agent exploration-exploitation tradeoff, which considers the balance of exploration and exploitation across multiple agents in addition to along the time dimension. To the best of our knowledge, our paper is the first work that investigates the multi-agent exploration-exploitation tradeoff and proposes an algorithm to handle this tradeoff.

We revised our paper based on the reviewers' comments.

    - To improve the clarity of the paper, we have added a detailed explanation of the computation of the proposed metric in Appendix B.1 and two separate overall structural diagrams for continuous-action and discrete-action cases in Appendix B.2.

    - We have added more baselines. In the cooperative continuous matrix game, we have included FOP as a baseline. For the SMAC environments, MAPPO has been included as a baseline.

    - We have added more experiments on the original SMAC environments and the Google Research Football environment, as the reviewers suggested. In both environments, it is shown that ADER significantly outperforms the baselines: QMIX, FOP, FACMAC, and SER-MARL. Due to the space limitation, this result has been included in Appendix D.

---

### Author Response · Authors · 2022-12-06
**Look forward to feedback**

We again thank all reviewers' valuable comments that helped us improve our paper.  We revised our paper based on the reviewers' comments including a detailed explanation for computing the proposed metric and further experiments. We hope our comments successfully address the reviewers' concerns.

Since the discussion period is coming to an end,  we look forward to discussing our paper.
Please let us know if you have any other concerns.

Thank you

Authors

---

### Decision · Program_Chairs · 2023-01-20

**Decision:**

Reject

**Justification For Why Not Higher Score:**

Missing comparison to MAPPO on original SMAC tasks.

**Justification For Why Not Lower Score:**

N/A

**Metareview: Summary, Strengths And Weaknesses:**


I thank the authors for their submission and active engagement during the discussion period.  This is a borderline paper with extensive internal discussion. On the positive side, the motivation of the method is convincing [5qy9,dyHK], the method is somewhat novel [5qy9,dyHK] and interesting [dyHK], and the paper is written clearly [5qy9] with a thorough literature review [mfnN], and overall and easy to read [dyHK]. On the negative side, reviewers noted experiments could have been more thorough [5qy9], for example demonstrating their method on more complex MARL benchmark [dyHK] as well as comparing to original tasks in the SMAC benchmark [dyHK,mfnN]. The authors have tried to address in particular the last two points in their rebuttal. However, they still have not provided a comparison against MAPPO on the original SMAC tasks. I side with reviewer mfnN on this issue and believe the paper requires a resubmission. I strongly encourage the authors to use the feedback by the reviewers to improve their paper.